# LEARNING TO INDUCE CAUSAL STRUCTURE

**Nan Rosemary Ke** [1,2,3], **Silvia Chiappa** [1], **Jane Wang** [1], **Anirudh Goyal** [1,2,4]
**Jorg Bornschein** [1], **Melanie Rey** [1], **Theophane Weber** [1], **Matthew Botvinick** [1]
**Michael Mozer** [5], **Danilo Jimenez Rezende** [1]

## ABSTRACT

One of the fundamental challenges in causal induction is to infer the underlying graph structure given observational and/or interventional data. Most existing causal induction algorithms operate by generating candidate graphs and evaluating them using either score-based methods (including continuous optimization) or independence tests. In our work, we instead treat the inference process as a black box and design a neural network architecture that learns the mapping from *both observational and interventional data* to graph structures via supervised training on synthetic graphs. The learned model generalizes to new synthetic graphs, is robust to train-test distribution shifts, and achieves state-of-the-art performance on naturalistic graphs for low sample complexity.

## 1 INTRODUCTION

The problem of discovering the causal relationships that govern a system through observing its behavior, either passively (*observational data*) or by manipulating some of its variables (*interventional data*), lies at the core of many important challenges spanning scientific disciplines, including medicine, biology, and economics. By using the graphical formalism of causal Bayesian networks (CBNs) (Koller & Friedman, 2009; Pearl, 2009), this problem can be framed as inducing the graph structure that best represents the relationships. Most approaches to causal structure induction (or causal discovery) are based on an unsupervised learning paradigm in which the structure is directly inferred from the system observations, either by ranking different structures according to some metrics (score-based approaches) or by determining the presence of an edge between pairs of variables using conditional independence tests (constraint-based approaches) (Drton & Maathuis, 2017; Glymour et al., 2019; Heinze-Deml et al., 2018a;b) (see Fig. 1(a)). The unsupervised paradigm poses however some challenges: score-based approaches are burdened with the high computational cost of having to explicitly consider all possible structures and with the difficulty of devising metrics that can balance goodness of fit with constraints for differentiating causal from purely statistical relationships (e.g. sparsity of the structure or simplicity of the generation mechanism); constraint-based methods are sensitive to failure of independence tests and require faithfulness, a property that does not hold in many real-world scenarios (Koski & Noble, 2012; Mabrouk et al., 2014).

Recently, supervised learning methods based on observational data have been introduced as an alternative to unsupervised approaches (Lopez-Paz et al., 2015a;b; Li et al., 2020). In this work, we extend the supervised learning paradigm to also use interventional data, enabling greater flexibility. We propose a model that is first trained on synthetic data generated using different CBNs to learn a mapping from data to graph structures and then used to induce the structures underlying datasets of interest (see Fig. 1(b)). The model is a novel variant of a transformer neural network that receives as input a dataset consisting of observational and interventional samples corresponding to the same CBN and outputs a prediction of the CBN graph structure. The mapping from the dataset to the underlying structure is achieved through an attention mechanism which alternates between attending to different variables in the graph and to different samples from a variable. The output is produced by a decoder mechanism that operates as an autoregressive generative model on the inferred structure. Our approach can be viewed as a form of meta-learning, whereby the relationship between datasets and causal structures underlying them are learned rather than built-in.

[0][1]DeepMind, [2] Mila, [3] Polytechnique Montreal, [4] University of Montreal, [5] Google Research, Brain Team, Corresponding author: nke@google.com

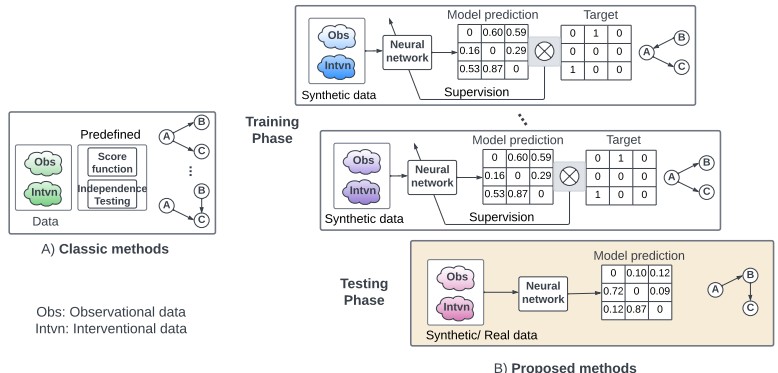

Figure 1: (A). Standard unsupervised approach to causal structure induction: Algorithms use a predefined scoring metric or statistical independence tests to select the best candidate structures. (B). Our supervised approach to causal structure induction (via attention; CSIvA): A model is presented with data and structures as training pairs and learns a mapping between them.

A requirement of a supervised approach would seem to be that the distributions of the training and test data match or highly overlap. Obtaining real-world training data with a known causal structure that matches test data from multiple domains is extremely challenging. We show that meta-learning enables the model to generalize well to data from naturalistic CBNs even if trained on synthetic data with relatively few assumptions, where naturalistic CBNs are graphs that correspond to causal relationships that exist in nature, such as graphs from the bnlearn repository (www.bnlearn.com/bnrepository). We show that our model can learn a mapping from datasets to structures and achieves state-of-the-art performance on classic benchmarks such as the Sachs, Asia and Child datasets (Lauritzen & Spiegelhalter, 1988; Sachs et al., 2005; Spiegelhalter & Cowell, 1992), despite never directly being trained on such data. Our contributions can be summarized as follows:

- We tackle causal structure induction with a supervised approach (CSIvA) that maps datasets composed of *both observational and interventional* samples to structures.

- We introduce a variant of a transformer architecture whose attention mechanism is structured to discover relationships among variables across samples.

- We show that CSIvA generalizes to novel structures, whether or not training and test distributions match. Most importantly, training on synthetic data transfers effectively to naturalistic CBNs.

- We show that CSIvA significantly outperforms state-of-the-art causal structure induction methods such as DCDI (Brouillard et al., 2020) and ENCO (Lippe et al., 2021) both on various types of synthetic CBNs, as well as on naturalistic CBNs.

## 2  BACKGROUND

In this section we give some background on causal Bayesian networks and transformer neural networks, which form the main ingredients of our approach (see Appendix A.1 for more details).

**Causal Bayesian networks (CBNs).**  A *causal Bayesian network* (Koller & Friedman, 2009; Pearl, 2009) is a pair $\mathcal{M} = \langle \mathcal{G}, p \rangle$, where $\mathcal{G}$ is a *directed acyclic graph* (DAG) whose nodes $X_1, \ldots, X_N$ represent random variables and edges express casual dependencies among them, and where $p$ is a joint distribution over all nodes such that $p(X_1, \ldots, X_N) = \prod_{n=1}^{N} p(X_n \mid \mathrm{pa}(X_n))$, where $\mathrm{pa}(X_n)$ are the *parents* of $X_n$, i.e. the nodes with an edge onto $X_n$ (direct causes of $X_n$). An input to the transformer neural network is formed by a dataset $\mathcal{D} = \{x^s\}_{s=1}^{S}$, where $x^s := (x_1^s, \ldots, x_N^s)^\mathsf{T}$ is either an *observational data sample* or an *interventional data sample* obtained by performing an intervention on a randomly selected node in $\mathcal{G}$. Observational data samples are samples from $p(X_1, \ldots, X_N)$. Except where otherwise noted, for all experimental settings, we considered *hard interventions* on a node $X_n$ that consist in replacing the conditional probability distribution (CPD) $p(X_n \mid \mathrm{pa}(X_n))$ with a delta function $\delta_{X_{n'}=x}$ forcing $X_{n'}$ to take on value $x$. Additional experiments were also performed using *soft interventions*, which consisted of replacing $p(X_n \mid \mathrm{pa}(X_n))$ with a different CPD $p'(X_n \mid \mathrm{pa}(X_n))$. An *interventional data sample* is a sample from $\delta_{X_{n'}=x} \prod_{n=1, n \neq n'}^{N} p(X_n \mid \mathrm{pa}(X_n))$ in the first case, and a sample from $p'(X_n \mid \mathrm{pa}(X_n)) \prod_{n=1, n \neq n'}^{N} p(X_n \mid \mathrm{pa}(X_n))$ in the second case. The structure of $\mathcal{G}$ can be repre-

sented by an adjacency matrix $A$, defined by setting the $(k, l)$ entry, $A_{k,l}$, to 1 if there is an edge from $X_l$ to $X_k$ and to 0 otherwise. Therefore, the $n$-th row of $A$, denoted by $A_{n,:}$, indicates the parents of $X_n$ while the $n$-th column, denoted by $A_{:,n}$, indicates the *children* of $X_n$.

**Transformer neural network.** A transformer (Devlin et al., 2018; Vaswani et al., 2017) is a neural network equipped with layers of self-attention that make them suited to modeling structured data. In traditional applications, attention is used to account for the sequentially ordered nature of the data, e.g. modeling a sentence as a stream of words. In our case, each input of the transformer is a dataset of observational or interventional samples corresponding to the same CBN. Attention is thus used to account for the structure induced by the CBN graph structure and by having different samples from the same node. Transformers are permutation invariant with respect to the positions of the input elements, ensuring that the graph structure prediction does not depend on node and sample position.

# 3 CAUSAL STRUCTURE INDUCTION VIA ATTENTION (CSIvA)

Our approach is to treat causal structure induction as a supervised learning problem, by training a neural network to learn to map *observational and interventional* data to the graph structure of the underlying CBN. Obtaining diverse, real-world data with known causal relationships in amounts sufficient for supervised training is not feasible. The key contribution of this work is to introduce a method that uses synthetic data generated from CBNs with different graph structures and associated CPDs that is robust to shifts between the training and test data distributions.

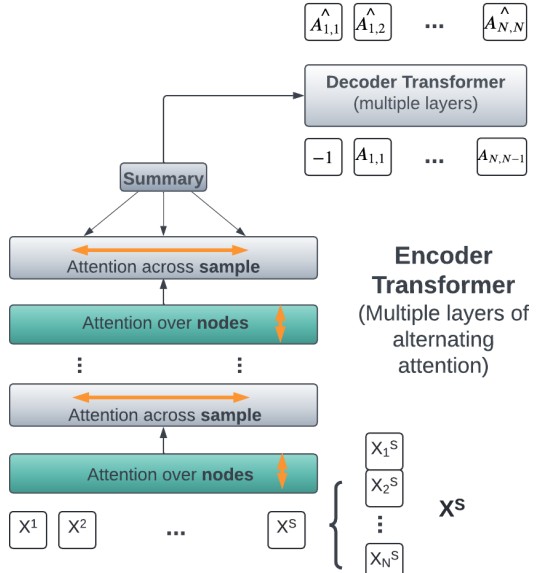

## 3.1 SUPERVISED APPROACH

We learn a distribution of graphs conditioned on observational and interventional data as follows. We generate training data from a joint distribution $t(\mathcal{G}, \mathcal{D})$, between a graph $\mathcal{G}$ and a dataset $\mathcal{D}$ comprising of $S$ *observational* and *interventional* samples from a CBN associated to $\mathcal{G}$ as follows. We first sample a set of graphs $\{\mathcal{G}^i\}_{i=1}^I$ with nodes $X_1^i, \ldots, X_N^i$ from a common distribution $t(\mathcal{G})$ as described in Section 4 (to simplify notation, in the remainder of the paper we omit the graph index $i$ when refer-

Figure 2: Our model architecture and the structure of the input and output at training time. The input is a dataset $\mathcal{D} = \{x^s := (x_1^s, \ldots, x_N^s)^\mathsf{T}\}_{s=1}^S$ of $S$ samples from a CBN and its adjacency matrix $A$. The output is a prediction $\hat{A}$ of $A$.

ring to nodes), and then associate random CPDs to the graphs as described in Section 4. This results in a set of CBNs $\{\mathcal{M}^i\}_{i=1}^I$. For each CBN $\mathcal{M}^i$, we then create a dataset $\mathcal{D}^i = \{x^s\}_{s=1}^S$, where each element $x^s := (x_1^s, \ldots, x_N^s)^\mathsf{T}$ is either an observational data sample or an *interventional* data sample obtained by performing an intervention on a randomly selected node in $\mathcal{G}^i$.

Our model defines a distribution $\hat{t}(\mathcal{G} \,|\, \mathcal{D}; \Theta)$ over graphs conditioned on observational and *interventional* data and parametrized by $\Theta$. Specifically, $\hat{t}(A \,|\, \mathcal{D}; \Theta)$ has the following auto-regressive form: $\hat{t}(A \,|\, \mathcal{D}; \Theta) = \prod_{l=1}^{N^2} \sigma(A_l; \hat{A}_l = f_\Theta(A_{1,\ldots,(l-1)}, \mathcal{D}))$, where $\sigma(\cdot; \rho)$ is the Bernoulli distribution with parameter $\rho$, which is a function $f_\Theta$ built from an encoder-decoder architecture explained in Section 3.2 taking as input previous elements of the adjacency matrix $A$ (represented here as an array of $N^2$ elements) and $\mathcal{D}$. It is trained via maximum likelihood estimation (MLE), i.e $\Theta^* = \operatorname{argmin}_\Theta \mathcal{L}(\Theta)$, where $\mathcal{L}(\Theta) = -\mathbb{E}_{(\mathcal{G}, \mathcal{D}) \sim t}[\ln \hat{t}(\mathcal{G} \,|\, \mathcal{D}; \Theta)]$, which corresponds to the usual cross-entropy (CE) loss for the Bernoulli distribution. Training is achieved using a stochastic gradient descent (SGD) approach in which each gradient update is performed using a pair $(\mathcal{D}^i, A^i)$. The data-sampling distribution $t(\mathcal{G}, \mathcal{D})$ and the MLE objective uniquely determine the target distribution learned by the model. In the infinite capacity case, $\hat{t}(\cdot \,|\, \mathcal{D}; \Theta^*) = t(\cdot \,|\, \mathcal{D})$. To see this, it suffices to note that the MLE objective $\mathcal{L}(\Theta)$ can be written as $\mathcal{L}(\Theta) = \mathbb{E}_{\mathcal{D} \sim t}[\text{KL}(\hat{t}(\cdot \,|\, \mathcal{D}; \Theta); t(\cdot \,|\, \mathcal{D}))] + c$, where KL is the Kullback-Leibler

divergence and $c$ is a constant. In the finite-capacity case, the distribution defined by the model $\hat{t}(\cdot \,|\, \mathcal{D}; \Theta^*)$ is only an approximation of $t(\cdot \,|\, \mathcal{D})$.

## 3.2 MODEL ARCHITECTURE

The function $f_\Theta$ defining the model's probabilities is built using two transformer networks. It is formed by an encoder transformer and by a decoder transformer (which we refer to as "encoder" and "decoder" for short). At training time, the encoder receives as input dataset $\mathcal{D}^i$ and outputs a representation that summarizes the relationship between nodes in $\mathcal{G}^i$. The decoder then recursively outputs predictions of the elements of the adjacency matrix $A^i$ using as input the elements previously predicted and the encoder output. This is shown in Fig. 2 (where we omitted index $i$, as in the remainder of the section). At test time we obtain deterministic predictions of the adjacency matrix elements by taking the argmax of the Bernoulli distribution for use as inputs to the decoder.

### 3.2.1 ENCODER

Our encoder is structured as an $(N + 1) \times (S + 1)$ lattice. The $N \times S$ part of the lattice formed by the first $N$ rows and first $S$ columns receives a dataset $\mathcal{D} = \{(x_1^s, \ldots, x_N^s)^\mathsf{T}\}_{s=1}^S$. This is unlike standard transformers which typically receive as input a single data sample (e.g., a sequence of words in neural machine translation applications) rather than a set of data samples. Row $N + 1$ of the lattice is used to specify whether each data sample is *observational*, through value $-1$, or *interventional*, through integer value in $\{1, \ldots, N\}$ to indicate the intervened node.

The goal of the encoder is to infer causal relationships between nodes by examining the set of samples. The transformer performs this inference in multiple stages, each represented by one transformer layer, such that each layer yields a $(N + 1) \times (S + 1)$ lattice of representations. The transformer is designed to deposit its summary representation of the causal structure in column $S + 1$.

**Embedding of the input.** Each data-sample element $x_n^s$ is embedded into a vector of dimensionality $H$. Half of this vector is allocated to embed the value $x_n^s$ itself, while the other half is allocated to embed the unique identity for the node $X_n$. The value embedding is obtained by passing $x_n^s$, whether discrete or continuous, through an MLPUsing an MLP for a discrete variable is a slightly inefficient implementation of a node value embedding, but it ensures that the architecture is general. encoder specific to node $X_n$. We use a node-specific embedding because the values of each node may have very different interpretations and meanings. The node identity embedding is obtained using a standard 1D transformer positional embedding over node indices. For column $S + 1$ of the input, the value embedding is a vector of zeros.

**Alternating attention.** Traditional transformers discover relationships among the elements of a data sample arranged in a one-dimensional sequence. With our two-dimensional lattice, the transformer could operate over the entire lattice at once to discover relationships among both nodes and samples. Given an encoding that indicates position $n, s$ in the lattice, the model can in principle discover stable relationships among nodes over samples. However, the inductive bias to encourage the model to leverage the lattice structure is weak. Additionally, the model is invariant to sample ordering, which is desirable because the samples are *iid*. Therefore, we arrange our transformer in alternating layers. In the first layer of the pair, attention operates across all nodes of a single sample $(x_1^s, \ldots, x_N^s)^\mathsf{T}$ to encode the relationships among two or more nodes. In the second layer of the pair, attention operates across all samples for a given node $(x_n^1, \ldots, x_n^S)$ to encode information about the distribution of node values. Alternating attention in transformers was also done in Kossen et al. (2021).

**Encoder summary.** The encoder produces a *summary* vector $e_n^{\text{sum}}$ with $H$ elements for each node $X_n$, which captures essential information about the node's behavior and its interactions with other nodes. The summary representation is formed independently for each node and involves combining information across the $S$ samples (the columns of the lattice). This is achieved with a method often used with transformers that involves a weighted average based on how informative each sample is. The weighting is obtained using the embeddings in column $S + 1$ to form queries, and embeddings in columns $1, \ldots, S$ to provide keys and values, and then using standard key-value attention.

### 3.2.2 DECODER

The decoder uses the summary information from the encoder to generate a prediction of the adjacency matrix $A$ of the underlying $\mathcal{G}$. It operates sequentially, at each step producing a binary output indicating the prediction $\hat{A}_{k,l}$ of $A_{k,l}$, proceeding row by row. The decoder is an autoregressive transformer, meaning that each prediction $\hat{A}_{kl}$ is obtained based on all elements of $A$ previously predicted, as well as the summary produced by the encoder. Our method does not enforce acyclicity. Although this could in principle yield cycles in the graph, in practice we observe strong performance regardless (Section 6.4), likely due to the fact that training and evaluation graphs typically studied (e.g., ER-1 and ER-2) are very sparse. Nevertheless, one could likely improve the results e.g. by using post-processing (Lippe et al., 2021) or by extending the method with an accept-reject algorithm (Castelletti & Mascaro, 2022; Li et al., 2022).

**Auxiliary loss.** Autoregressive decoding of the flattened $N \times N$ adjacency matrix can be difficult for the decoder to learn alone. To provide additional inductive bias to facilitate learning, we added auxiliary task of predicting the parents $A_{n,:}$ and children $A_{:,n}$ of node $X_n$ from the encoder summary, $e_n^{\text{sum}}$. This is achieved using an MLP to learn a mapping $f_n$, such that $f_n(e_n^{\text{sum}}) = (\hat{A}_{n,:}, \hat{A}_{:,n}^{\mathsf{T}})$. While this prediction is redundant with the operation of the decoder, it short-circuits the autoregressive decoder and provides a strong training signal to support proper training.

## 4 SYNTHETIC DATA

In this section, we discuss identifiability and describe how synthetic data were generated.

**Identifiability.** Dataset $\mathcal{D}^i$ associated to CBN $\mathcal{M}^i$ is given by $\mathcal{D}^i = \{x^s\}_{s=1}^S$, where $x^s := (x_1^s, \ldots, x_N^s)^{\mathsf{T}}$ is either an observational or interventional data sample obtained by performing a hard intervention on a randomly selected node in $\mathcal{G}^i$. As discussed in Eberhardt et al. (2006), in the limit of an infinite amount of such single-node interventional data samples, $\mathcal{G}^i$ is identifiable. Note, that identifiability here means the ability to recover the exact graph given the data. As our model defines a distribution over graphs, its predictions are meaningful even when the amount of data is insufficient for identifiability: in this case, the model would sample graphs that are compatible with the given data. Empirically, we found that our model can make reasonable predictions even with a small amount of samples per intervention and improves as more samples are observed (Section 6.4).

**Graph distribution.** We specified a distribution over $\mathcal{G}$ in terms of the number of nodes $N$ (graph size) and number of edges (graph density) present in $\mathcal{G}$. As shown in Zheng et al. (2018); Yu et al. (2019); Ke et al. (2020a), larger and denser graphs are more challenging to learn. We varied $N$ from $5$ to $80$. We used the Erdős–Rényi (ER) and the scale-free (SF) metrics to vary density and evaluated our model on ER-1 and ER-2 graphs, as in Yu et al. (2019); Brouillard et al. (2020); Scherrer et al. (2021). We generated an adjacency matrix $A$ by first sampling a lower-triangular matrix to ensure that it represents a DAG, and by then permuting the order of the nodes to ensure random ordering.

**Conditional probability distributions.** We performed ancestral sampling on the underlying CBN. We considered both continuous and discrete nodes. For continuous nodes, we generated *continuous data* using three methods following similar setups in previous works: (i) linear models (*linear data*) (Zheng et al., 2018; Yu et al., 2019), (ii) nonlinear with additive noise models (*ANM data*) (Brouillard et al., 2020; Lippe et al., 2021), and (iii) nonlinear with non-additive noise models using neural networks *(NN data)* (Brouillard et al., 2020; Lippe et al., 2021). For discrete nodes, we generated discrete data using two different methods: MLP (*MLP data*) and Dirichlet (*Dirichlet data*) conditional-probability table generators. Following past work (Ke et al., 2020a; Scherrer et al., 2021), we used a randomly initialized network. The Dirichlet generator filled in the rows of a conditional probability table by sampling a categorical distribution from a Dirichlet prior with symmetric parameters $\alpha$. (We remind the reader that this generative procedure is performed prior to node ordering being randomized for presentation to the learning model.) Refer to Appendix A.2 for details.

## 5 RELATED WORK

Methods for inferring causal graphs can broadly be categorized into score-based, constraint-based, and asymmetry-based methods. Score-based methods search through the space of possible candidate graphs, and ranks them based on some scoring function (Chickering, 2002; Cooper & Yoo, 1999;

Goudet et al., 2017; Hauser & Bühlmann, 2012; Heckerman et al., 1995; Tsamardinos et al., 2006; Huang et al., 2018; Zhu et al., 2019). Recently, Zheng et al. (2018); Yu et al. (2019); Lachapelle et al. (2019) framed the structure search as a continuous optimization problem. There exist score-based methods that use a mix of continuous and discrete optimization (Bengio et al., 2019; Zhu et al., 2019; Ke et al., 2020a; Lippe et al., 2021; Scherrer et al., 2021). Constraint-based methods (Monti et al., 2019; Spirtes et al., 2000; Sun et al., 2007; Zhang et al., 2012; Zhu et al., 2019) infer the DAG by analyzing conditional independencies in the data. Eaton & Murphy (2007) use dynamic programming techniques. Asymmetry-based methods (Shimizu et al., 2006; Hoyer et al., 2009; Peters et al., 2011; Daniusis et al., 2012; Budhathoki & Vreeken, 2017; Mitrovic et al., 2018) uses asymmetry between cause and effect to estimate the causal structure. Peters et al. (2016); Ghassami et al. (2017); Rojas-Carulla et al. (2018); Heinze-Deml et al. (2018a) exploit invariance across environments. Mooij et al. (2016) propose a modeling framework that leverages existing methods.

Learning-based methods have been proposed (Bengio et al., 2019; Goudet et al., 2018; Guyon, 2013; 2014; Kalainathan et al., 2018; Ke et al., 2020a;b; Lachapelle et al., 2022; Lopez-Paz et al., 2015b; Wang et al., 2021b; Zhu et al., 2019). In particular, Zhu et al. (2019); Wang et al. (2021b) use transformers. These works are concerned with learning only part of the causal induction pipeline, such as

| Data Type | RCC | DAG-EQ | CSIvA |
|---|---|---|---|
| Observational | ✓ | ✓ | ✓ |
| Interventional | X | X | ✓ |
| Linear dependencies | ✓ | ✓ | ✓ |
| Non-linear dependencies | ✓ | X | ✓ |

Table 1: Data-type comparison between CSIvA and other supervised approaches to causal structure induction (RCC (Lopez-Paz et al., 2015a;b) and DAG-EQ (Li et al., 2020)).

the scoring function, and hence are significantly different from our work, which uses an end-to-end supervised learning approach to learn to map from datasets to graphs. Neural network methods equipped with learned masks exist (Douglas et al., 2017; Goyal et al., 2021; Ivanov et al., 2018; Li et al., 2019; Yoon et al., 2018), but only a few have been adapted to causal inference. Several transformer models (Goyal et al., 2022; Kossen et al., 2021; Müller et al., 2021) have been proposed for learning to map from datasets to targets. However, none have been applied to causal discovery, although Löwe et al. (2022) proposes a neural-network based approach for causal discovery on time-series data. A few supervised learning approaches have been proposed either framing the task as a kernel mean embedding classification problem (Lopez-Paz et al., 2015a;b) or operating directly on covariance matrices (Li et al., 2020) or binary classification problem of identifying v-structures (Dai et al., 2021). These models accept observational data only (see Table 1), and because causal identifiability requires *both observational and interventional* data, our model is in principle more powerful.

## 6 EXPERIMENTS

We report on a series of experiments of increasing challenge to our supervised approach to causal structure induction. First, we examined whether CSIvA generalizes well on synthetic data for which the training and test distributions are identical (Section 6.1). This experiment tests whether the model can learn to map from a dataset to a structure. Second, we examined generalization to an out-of-distribution (OOD) test distribution, and we determined hyperparameters of the synthetic data generating process that are most robust to OOD testing (Section 6.2). Third, we trained CSIvA using the hyperparameters from our second experiment and evaluated it on a different type of OOD test distribution from several naturalistic CBNs (Section 6.3). This experiment is the most important test of our hypothesis that causal structure of synthetic datasets can be a useful proxy for discovering causal structure in realistic settings. Lastly, we performed a set of ablation studies to analyze the performance of CSIvA under different settings (Section 6.4). All models are trained for $500k$ iterations, please refer to Appendix A.3 for details on the hyperparameters.

**Comparisons to baselines.** We compared CSIvA to a range of methods considered to be state-of-the-art in the literature, ranging from classic to neural-network based causal discovery baselines. For both in-distribution and OOD experiments, we compare to 4 very strong baselines: DAG-GNN (Yu et al., 2019), non-linear ICP (Heinze-Deml et al., 2018b), DCDI (Brouillard et al., 2020), and ENCO (Lippe et al., 2021). For OOD experiments to naturalistic graphs, we compare to 5 additional baselines (Chickering, 2002; Hauser & Bühlmann, 2012; Zheng et al., 2018; Gamella & Heinze-Deml, 2020;

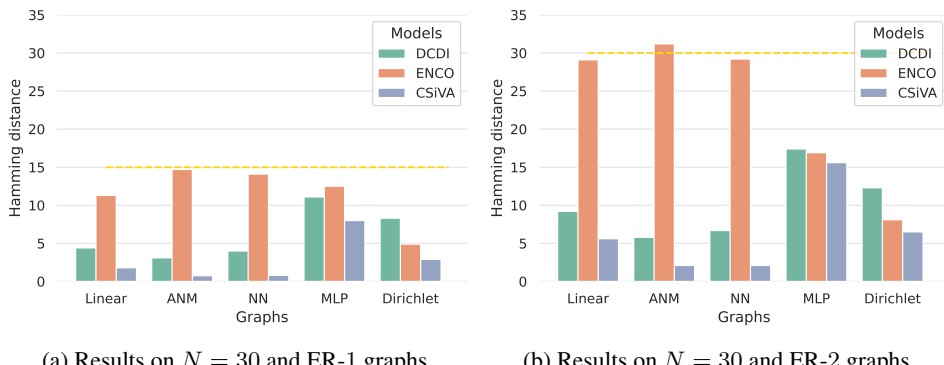

(a) Results on $N = 30$ and ER-1 graphs.      (b) Results on $N = 30$ and ER-2 graphs.

Figure 4: Hamming distance $\mathcal{H}$ between predicted and ground-truth adjacency matrices on 5 different data types: Linear, ANM, NN, MLP and Dirichlet data, compared to DCDI (Brouillard et al., 2020) and ENCO (Lippe et al., 2021) on $N = 30$ graphs, averaged over 128 sampled graphs. The dotted line indicates the value of the all-absent baseline. CSIvA significantly outperforms all other baselines for all data types.

Li et al., 2020). Because some methods use only observational data and others do not scale well to large N, we could not compare to all methods in all experiments. See Appendix A.4 for further discussion of alternative methods and conditions under which they can be used.

## 6.1 IN-DISTRIBUTION EXPERIMENTS

We begin by investigating whether CSIvA can learn to map from data to structures in the case in which the training and test distributions are identical. In this setting, our supervised approach has an advantage over unsupervised ones, as it can learn about the training distribution and leverage this knowledge during testing. We evaluate CSIvA for $N \leq 80$ graphs.

We examined the performance on data with increasing order of difficulty, starting with linear (continuous data), before moving to non-linear cases (ANM, NN, MLP and Dirichlet data). See Fig. 4 for comparisons between CSIvA and strong baselines on all data types ($N = 30$), showing that CSIvA significantly outperforms baselines for a wide range of data types. For all but the Dirichlet data, we summarize the high-level results here, but detailed results and plots for all experiments can be found in Appendix A.5.

Results on continuous linear data are presented in Tables 4, 5 and results on continuous ANM data are presented in Table 6, results on continuous NN data are reported in Table 7 in Appendix. Results on MLP data are shown in Fig. 5(b) (Appendix). For all data types and all graphs, CSIvA significantly outperforms non-linear ICP, DAG-GNN, DCDI and ENCO. Differences become more apparent with larger graph sizes ($N \geq 10$) and denser graphs (ER-2 vs ER-1).

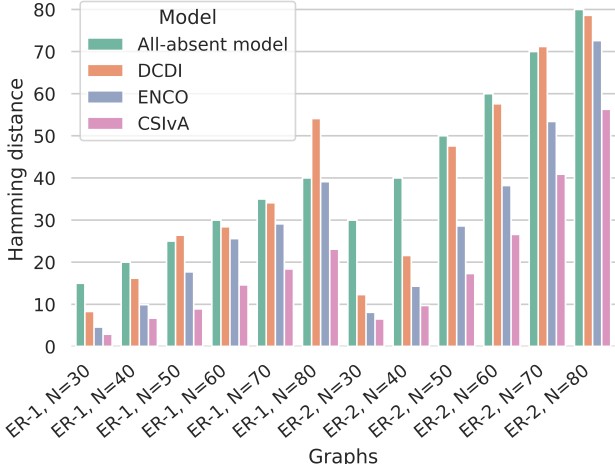

Figure 3: **Results on Dirichlet data for** $30 \leq N \leq 80$. Hamming distance $\mathcal{H}$ between predicted and ground-truth adjacency matrices, averaged over 128 sampled graphs. CSIvA significantly outperforms DCDI and ENCO, both of which are very strong baselines. The difference in performance increases with $N$.

The Dirichlet data requires setting the values of the parameter $\alpha$. Hence, we run two sets of experiments on this data. In the first set of experiments, we investigated how different values of $\alpha$ impact learning in CSIvA. As shown in Table 10 in the appendix, CSIvA performs well on all data with $\alpha \leq 0.5$, achieving $\mathcal{H} < 2.5$ in all cases. CSIvA still performs well when $\alpha = 1.0$, achieving

$\mathcal{H} < 5$ on size 10 graphs. Learning with $\alpha > 1$ is more challenging. This is not surprising, as $\alpha > 1$ tends to generate more uniform distributions, which are not informative of the causal relationship between nodes. In the second set of experiments, we compared CSIvA to DCDI and ENCO, as they are the strongest performing baselines. We run the comparisons on graphs of size up to $N \leq 80$. To limit the number of experiments to run, we set $\alpha = 0.1$, as this allows the conditional probabilities to concentrate on non-uniform distributions, which is more likely to be true in realistic settings. As shown in Fig. 3, our model significantly outperforms DCDI and ENCO on all graphs (up to $N \leq 80$). As with other data types, the difference becomes more apparent as the size of the graphs grows ($N \geq 50$), as larger graphs are more challenging to learn.

**Soft interventions and unknown interventions.** We additionally evaluated CSIvA on soft interventions, and on hard interventions for which the model did not know the intervention node (*unknown interventions*). In both cases, we focus on Dirichlet data with $\alpha = 0.1$ and $\alpha = 0.25$ for ER-1 and ER-2 graphs of size $N \leq 20$. Results for the performance of CSIvA on soft interventions are shown in Table 13 in the Appendix. CSIvA performed well on all graphs, achieving Hamming distance $\mathcal{H} < 5$ on all graphs. For details about the data generation process, refer to Appendix Section A.5.7.

Results for the performance of CSIvA on unknown interventions are shown in Appendix Table 16. CSIvA achieves strong performance on all graphs, with Hamming distance $\mathcal{H} < 7$ on all graphs. Note that CSIvA loses almost no performance despite not knowing the intervention; the biggest difference in Hamming distance between known and unknown interventions on all graphs is $\mathcal{H} < 1.0$. For more details, refer to Section A.5.8 in the Appendix.

## 6.2 OUT-OF-DISTRIBUTION EXPERIMENTS

In this set of experiments, we evaluated CSIvA's ability to generalize with respect to aspects of the data generating distribution that are often unknown, namely graph density and parameters of the CPDs, such as the $\alpha$ values of the Dirichlet distribution.

**Varying graph density.** We evaluated how well our model performs when trained and tested on CBNs with varying graph density on MLP and $\alpha = 1$ Dirichlet data and the number of nodes to $N = 7$, with variables able to take on discrete values in $\{1, 2, 3\}$. The graphs in training and test datasets can take ER degree $\in \{1, 2, 3\}$. Results are shown in Table 17 (Appendix) for the MLP data and Table 18 (Appendix) for the Dirichlet data. For the MLP data, models trained on ER-2 graph generalizes the best. For Dirichlet data, there isn't one value of graph density that consistently generalizes best across graphs with different densities. Nevertheless, ER-2 graphs give a balanced trade-off and generalizes well across graphs with different sparsity.

**Varying $\alpha$.** We next trained and evaluated on data generated from Dirichlet distributions with $\alpha \in \{0.1, 0.25, 0.5\}$. Results for ER-1 graphs with $N = 7$ are found in Table 19. There isn't a value of $\alpha$ that performs consistently well across different values of $\alpha$ for the test data. Nevertheless, $\alpha = 0.25$ is a balanced trade-off and generalizes well across test data with $0.1 \leq \alpha \leq 0.5$.

## 6.3 SIM-TO-REAL EXPERIMENTS

In this final set of experiments, we evaluated CSIvA's ability to generalize from being trained on MLP and Dirichlet data to being evaluated on the widely used Sachs (Sachs et al., 2005), Asia (Lauritzen & Spiegelhalter, 1988) and Child (Spiegelhalter & Cowell, 1992) CBNs from the bnlearn repository, which have $N = 11$, $N = 8$ and $N = 20$ nodes respectively. We followed the established protocol from Ke et al. (2020a); Lippe et al. (2021); Scherrer et al. (2021) of sampling observational and interventional data from the CBNs provided by the repository. These experiments are the most important test of our hypothesis that causal structure of synthetic datasets can be a useful proxy for discovering causal structure in realistic settings.

We emphasize that all hyperparameters for the MLP and Dirichlet data and for the learning procedure were chosen *a priori*; only after the architecture and parameters were finalized was the model tested on these benchmarks. Furthermore, to keep the setup simple, we trained on data sampled from a single set of hyperparameters instead of a broader mixture. Findings in Section 6.2 suggest that ER-2 graphs with $\alpha = 0.25$ work well overall and hence were chosen. Results are reported in Table 2, comparing to a range of baselines from Squires et al. (2020); Heinze-Deml et al. (2018b); Yu et al. (2019); Gamella & Heinze-Deml (2020); Brouillard et al. (2020); Lippe et al. (2021) and others.

CSIvA trained on both MLP and Dirichlet data significantly outperforms all other methods on both the Asia and Sachs datasets. This serves as strong evidence that our model can learn to induce causal structures in the more realistic real-world CBNs, while only trained on synthetic data.

### 6.4 DETAILED ANALYSES

In order to better understand model performance, and which model components were most critical, we performed further analyses. We describe here a subset of the studies performed; please refer to Appendix A.7 for the full results, including ablations.

|  | Asia | Sachs | Child |
|---|---|---|---|
| Number of nodes | 8 | 11 | 20 |
| All-absent Baseline | 8 | 17 | 25 |
| GES (Chickering, 2002) | 4 | 19 | 33 |
| DAG-notears (Zheng et al., 2018) | 14 | 22 | 23 |
| DAG-GNN (Yu et al., 2019) | 8 | 13 | 20 |
| GES (Hauser & Bühlmann, 2012) | 11 | 16 | 31 |
| ICP (Peters et al., 2016) | 8 | 17 | 27* |
| Non-linear ICP (Heinze-Deml et al., 2018b) | 8 | 16 | 23* |
| DAG-EQ (Li et al., 2020) | - | 16 | - |
| DCDI-DSF (Brouillard et al., 2020) | 7 | 33 | 18 |
| ENCO (Lippe et al., 2021) | 5 | 9 | 14 |
| CSIvA (MLP data) | **3** | **6** | **11** |
| CSIvA  (Dirichlet data) | **3** | **5** | **10** |

Table 2: **Results on Asia, Sachs and Child data:** Hamming distance $\mathcal{H}$ between predicted and ground-truth adjacency matrices. *To maintain computational tractability, the number of parents considered was limited to 3.

**Acyclicity.** We analyzed the generated (sampled) graphs for acyclicity under several settings. We found that none of the generated graphs contains any cycles. For details, refer to Appendix A.7.3.

**Visualization of generated graphs.** We visualized some of the generated graphs in Figure 7 and Figure 8 in the Appendix. Note that samples are randomly drawn, and all are acyclic.

**Identifiability upon seeing intervention data.** Here, we investigate how varying the proportion of interventional data impacts the performance of the proposed model. As shown in Figure 9 (Appendix), the model's performance improves (Hamming distance decreases) almost monotonically as the amount of interventional data increases, from $0 - 100\%$. This is a clear indication that our model is able to extract information from interventional data for identifying the graph structure. For more details on the results and the how the experiments are conducted, please refer to Section A.7.4.

**Amount of training cases.** We conducted experiments to better understand how much synthetic data is needed for different graph sizes. For smaller graphs ($N \leq 25$), there is a small improvement for using more than $10k$ training cases. For larger graphs ($N > 25$) the performance consistently improves as the model uses more training data (more details in Appendix A.7.6).

**Ablation studies.** We investigate the role of various components in CSIvA. We found that all the different components play an important role in our model. For details, see Appendix A.7.

**Computation time.** Our models are trained for $500k$ iterations. Training can take up to 8 hours, and then inference takes minutes. Training time of CSIvA can be amortized for multiple test cases; training time for baseline models are proportional to the number of test cases. For more details, please refer to Appendix A.7.9.)

## 7 DISCUSSION

In this paper, we have presented a novel approach towards causal graph structure inference. Our method is based on learning from synthetic data in order to obtain a strong learning signal (in the form of explicit supervision), using a novel transformer-based architecture which directly analyzes the data and computes a distribution of candidate graphs. Through a comprehensive and detailed set of experiments, we demonstrated that even though only trained on synthetic data, our model generalizes to out-of-distribution datasets, and robustly outperforms comparison methods under a wide range of conditions. A direction of future work would be to use the proposed framework for learning causal structure from raw visual data. This could be useful, e.g. in an RL setting in which an RL agent interacts with the environment via observing low level pixel data (Ahmed et al., 2020; Ke et al., 2021; Wang et al., 2021a).

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

# A  APPENDIX

## CONTENTS

## A.1 TRANSFORMER NEURAL NETWORKS

The transformer architecture, introduced in Vaswani et al. (2017), is a multi-layer neural network architecture using stacked self-attention and point-wise, fully connected, layers. The classic transformer architecture has an encoder and a decoder, but the encoder and decoder do not necessarily have to be used together.

**Scaled dot-product attention.** The attention mechanism lies at the core of the transformer architecture. The transformer architecture uses a special form of attention, called the scaled dot-product attention. The attention mechanism allows the model to flexibility learn to weigh the inputs depending on the context. The input to the QKV attention consists of a set of queries, keys and value vectors. The queries and keys have the same dimensionality of $d_k$, and values often have a different dimensionality of $d_v$. Transformers compute the dot products of the query with all keys, divide each by $\sqrt{d_k}$, and apply a softmax function to obtain the weights on the values. In practice, transformers compute the attention function on a set of queries simultaneously, packed together into a matrix $Q$. The keys and values are also packed together into matrices $K$ and $V$. The matrix of outputs is computed as: $\text{Attention}(Q, K, V) = \text{softmax}(\frac{QK^T}{\sqrt{d_k}})V$.

**Encoder.** The encoder is responsible for processing and summarizing the information in the inputs. The encoder is composed of a stack of $N$ identical layers, where each layer has two sub-layers. The first sub-layer consists of a multi-head self-attention mechanism, and the second is a simple, position-wise fully connected feed-forward network. Transformers employ a residual connection (He et al., 2016) around each of the two sub-layers, followed by layer normalization (Ba et al., 2016). That is, the output of each sub-layer is $\text{LayerNorm}(x + \text{Sublayer}(x))$, where $\text{Sublayer}(x)$ is the function implemented by the sub-layer itself.

**Decoder.** The decoder is responsible for transforming the information summarized by the encoder into the outputs. The decoder also composes of a stack of $N$ identical layers, with a small difference in the decoder transformer layer. In addition to the two sub-layers in each encoder layer, a decoder layer consists of a third sub-layer. The third sub-layer performs a multi-head attention over the output of the encoder stack. Similar to the encoder, transformers employ residual connections around each of the sub-layers, followed by layer normalization. Transformers also modify the self-attention sub-layer in the decoder stack to prevent positions from attending to subsequent positions. This masking, combined with fact that the output embeddings are offset by one position, ensures that the predictions for position $i$ can depend only on the known outputs at positions less than $i$.

## A.2 SYNTHETIC DATA

In Section 4, we introduced the types of methods used to generate the conditional probability distribution. In this section, we discuss the details of these methods.

We evaluate our model on datasets generated from 5 different methods, which covers both continuous and discrete valued data generated with a varying degree of difficulty. For continuous data, we generated the data using three different methods: *linear*, *non-linear additive noise model* (ANM) and *non-linear non-additive noise neural network model* (NN). For discrete data, we generated data using two different methods: *MLP* and *Dirichlet*. Let $X$ be a $N \times S$ matrix representing $S$ samples of a CBN with $N$ nodes and weighted adjacency matrix $A$, and let $Z$ be a random matrix of elements in $\mathcal{N}(0, 0.1)$. We describe each type of how different types of data is generated.

- For *linear data*, we follow the setup in Zheng et al. (2018) and Yu et al. (2019). Specifically, we generated data as $X_{n,:} = A_{n,:}X + Z_{n,:}$. The biases were initialized using $U[-0.5, 0.5]$, and the individual weights were initialized using a truncated normal distribution with standard deviation of $1.5$. For nodes with interventions, values are sampled from the uniform distribution $U[-1, 1]$.

- For *additive noise models (ANM)*, we follow the setup in (Brouillard et al., 2020; Lippe et al., 2021). We generated the data as $X_{n,:} = F_{n,:}(X) + 0.4 \cdot Z_{n,:}$, where $F$ is fully connected neural network, the weights are randomly initialized from $\mathcal{N}(0, 1)$. The neural network has one hidden layer with 10 hidden units, the activation function is a leaky relu

with a slop of $0.25$. The noise variables are sampled from $\mathcal{N}(0, \sigma^2)$, where $\sigma^2 \sim \mathcal{U}[1, 2]$. For nodes with interventions, values are sampled from the uniform distribution $\mathcal{N}(2, 1)$.

- For *non-additive noise neural network (NN) models* , we also follow the setup in (Brouillard et al., 2020; Lippe et al., 2021). We generate the data as $X_{n,:} = F_{n,:}(X, Z_{n,:})$, such that $F$ is fully connected neural network, the weights are randomly initialized from $\mathcal{N}(0, 1)$. The neural network has one hidden layer with 20 hidden units, the activation function is a tanh function. The noise variables are sampled from $\mathcal{N}(0, \sigma^2)$, where $\sigma^2 \sim \mathcal{U}[1, 2]$. For nodes with interventions, values are sampled from the uniform distribution $\mathcal{N}(2, 1)$.

- For *MLP data*, the neural network has two fully connected layers of hidden dimensionality 32. Following past work (Ke et al., 2020a; Scherrer et al., 2021; Lippe et al., 2021), we used a randomly initialized network. For nodes with interventions, values are randomly and independently sampled from $U\{1, \ldots, K\}$ where $K$ indicates the number of categories of the discrete variable.

- For Dirichlet data, the generator filled in the rows of a conditional probability table by sampling a categorical distribution from a Dirichlet prior with symmetric parameters $\alpha$. Values of $\alpha$ smaller than 1 encourage lower entropy distributions; values of $\alpha$ greater than 1 provide less information about the causal relationships among variables. Similar to MLP data, for nodes with interventions, values are randomly and independently sampled from $U\{1, \ldots, K\}$ where $K$ indicates the number of categories of the discrete variable.

## A.3 HYPERPARAMETERS

Table 3: Hyperparameters used in all of our experiments.

| Hyperparameter | Value |
| --- | --- |
| Hidden state dimension | 64 |
| Encoder transformer layers | 8 |
| Num. attention heads | 8 |
| Optimizer | Adam |
| Learning rate | $10^{-4}$ |
| Number of random seeds | 3 |
| $S$ (number of samples) | 1500 |
| Training iterations | $500,000$ |
| Num. training datasets $I$ | $15,000$ ($N \leq 20$) $20,000$ ($N > 20$) |

For all of our experiments (unless otherwise stated) our model was trained on $I = 15,000$ (for graphs $N \leq 20$) and on $I = 20,000$ (for graphs $N > 20$) pairs $\{(\mathcal{D}^i, A^i)\}_{i=1}^I$, where each dataset $\mathcal{D}^i$ contained $S = 1500$ observational and interventional samples. For experiments on discrete data, a data-sample element $x^s$ could take values in $\{1, 2, 3\}$. Details of the data generating process can be found in Section 4. For evaluation in Sections 6.1 and 6.2, our model was tested on $I' = 128$ (different for the training) pairs $\{(\mathcal{D}^{i'}, A^{i'})\}_{i'=1}^{I'}$, where each dataset $\mathcal{D}^{i'}$ contained $S = 1500$ observational and *interventional* samples. For the Asia, Sachs and Child benchmarks, our model was still tested on $I' = 128$ (different for the training) pairs $\{(\mathcal{D}^{i'}, A^{i'})\}_{i'=1}^{I'}$, however, $A^{i'} = A^{j'}$ since there is only a single adjacency matrix in each one of the benchmarks. We present test results averaging performance over the 128 datasets and 3 random seeds and up to size $N = 80$ graphs. The model was trained for $500,000$ iterations using the Adam optimizer (Kingma & Ba, 2014) with a learning rate of $1e-4$. Also, refer to Table 3 for the list of hyperparameters presented in a table.

We parameterized our architecture such that inputs to the encoder were embedded into 128-dimensional vectors. The encoder transformer has 10 layers and 8 attention-heads per layer. The final attention step for summarization has 8 attention heads. The decoder was a smaller transformer with only 2 layers and 8 attention heads per layer. Discrete inputs were encoded using an embedding layer before passing into our model.

### A.4 COMPARISONS TO BASELINES

In Section 6.1, we compare CSIvA to four strong baselines in the literature, ranging from classic causal discovery baselines to neural-network based causal discovery baselines. These baselines are DAG-GNN (Yu et al., 2019), non-linear ICP (Heinze-Deml et al., 2018b), DCDI (Brouillard et al., 2020) and ENCO (Lippe et al., 2021).

Non-linear ICP, DCDI and ENCO can handle both observational and *interventional* data, while DAG-GNN can only use observational data. Non-linear ICP could not scale to graphs larger than 20, therefore we compare to DCDI (Brouillard et al., 2020) and ENCO (Lippe et al., 2021) on larger graphs. All baselines are unsupervised methods, i.e. they are not tuned to a particular training dataset but instead rely on a general-purpose algorithm. We also compared to an all-absent model corresponding to a zero adjacency matrix, which acts as a sanity check baseline. We also considered other methods (Chickering, 2002; Hauser & Bühlmann, 2012; Zhang et al., 2012; Gamella & Heinze-Deml, 2020), but only presented a comparison with DCDI, ENCO, non-linear ICP and DAG-GNN as these have shown to be strong performing models in other works (Ke et al., 2020a; Lippe et al., 2021; Scherrer et al., 2021).

For Section 6.3, we also compared to additional baselines from Chickering (2002); Hauser & Bühlmann (2012); Zheng et al. (2018); Gamella & Heinze-Deml (2020); Li et al. (2020). Note that methods from Chickering (2002); Zheng et al. (2018); Yu et al. (2019); Li et al. (2020) take observational data only. DAG-GNN outputs several candidate graphs based on different scores, such as evidence lower bound or negative log likelihood, DCDI can also be run in two different settings (DCDI-G and DCDI-DSF), we chose the best result to compare to our model. Note that non-linear ICP does not work on discrete data, i.e. on the MLP and Dirichlet data, therefore a small amount of Gaussian noise $\mathcal{N}(0, 0.1)$ was added to this data in order for the method to run.

### A.5 DETAILED RESULTS FOR SECTION 6.1

We present detailed results for experiments in Section 6.1 are described in the tables below.

### A.5.1 RESULTS ON LINEAR DATA

| | ER = 1 | | | | ER = 2 | | | |
|---|---|---|---|---|---|---|---|---|
| | Var = 5 | Var = 10 | Var = 15 | Var = 20 | Var = 5 | Var = 10 | Var = 15 | Var = 20 |
| Abs. | 2.50 | 5.00 | 7.50 | 10.00 | 5.00 | 10.00 | 15.00 | 20.00 |
| DAG-GNN | 2.71 | 4.76 | 7.71 | 11.32 | 5.20 | 8.81 | 17.81 | 22.21 |
| Non-linear ICP | 0.47 | 1.10 | 6.3 | 8.6 | 0.90 | 2.41 | 13.52 | 17.71 |
| DCDI | 0.23 | 0.78 | 4.51 | 6.21 | 0.92 | 2.01 | 8.12 | 12.59 |
| ENCO | 0.19 | 0.45 | 3.12 | 4.23 | 0.85 | 1.81 | 6.14 | 8.45 |
| Our Model | $\mathbf{0.12} \pm 0.03$ | $\mathbf{0.35} \pm 0.05$ | $\mathbf{2.01} \pm 0.07$ | $\mathbf{2.21} \pm 0.07$ | $\mathbf{0.81} \pm 0.05$ | $\mathbf{1.73} \pm 0.04$ | $\mathbf{5.32} \pm 0.19$ | $\mathbf{5.86} \pm 0.21$ |

Table 4: **Results on Continuous linear data**. Hamming distance $\mathcal{H}$ for learned and ground-truth edges on synthetic graphs, compared to other methods, averaged over 128 sampled graphs. The number of variables varies from 5 to 20, expected degree = 1 or 2, and the value of variables are drawn from $\mathcal{N}(0, 0.1)$. Note that for (Heinze-Deml et al., 2018b), the method required nodes to be causally ordered, and 20 repeated samples taken per intervention, as interventions were continuously valued. "Abs" baselines are All-Absent baselines, which is an baseline model that outputs all zero edges for the adjacency matrix. DCDI and ENCO are the strongest performing baselines. CSIvA outperforms all baselines (including DCDI and ENCO).

Results for comparisons between our model CSIvA and baselines non-linear ICP (Heinze-Deml et al., 2018b), DAG-GNN (Yu et al., 2019), DCDI (Brouillard et al., 2020) and ENCO (Lippe et al., 2021) are shown in Table 4 for smaller graphs with $N \leq 20$ and Table 5 for larger graphs with $20 < N \leq 80$.

For smaller graphs with $N \leq 20$, all models that takes interventional data perform significantly better compared to DAG-GNN (Yu et al., 2019), which only takes observational data. CSIvA achieves Hamming distance $\mathcal{H} < 7$ on evaluated graphs up to size 20. Similar to previous findings (Yu et al.,

2019; Ke et al., 2020a), larger and denser graphs are more challenging to learn. Non-linear ICP achieves fairly good performance on smaller graphs ( $N \leq 10$), however, the performance drops quickly as size of graphs increases ($N > 10$). Also, note that Non-linear ICP can not scale to graphs larger than $N > 20$. It also required a modificationWithout this modification, the method achieved near chance performance. to the dataset wherein multiple samples were collected from the same modified graph after a point intervention (20 samples per intervention), while other methods only sampled once per intervention.

For larger graphs of $20 < N \leq 280$, we compare to strongest baselines: DCDI and ENCO. Results are found in Table 5. CSIvA significantly outperforms both DCDI and ENCO. The difference because more apparent as the size of the graphs grow ($N \geq 50$).

| Dataset | All-absent | DCDI (Brouillard et al., 2020) | ENCO (Lippe et al., 2021) | CSIvA |
|---------|------------|-------------------------------|---------------------------|-------|
| $N = 30, ER = 1$ | 15.0 | 4.4 | 11.3 | **1.8** |
| $N = 50, ER = 1$ | 25.0 | 18.1 | 27.5 | **3.5** |
| $N = 80, ER = 1$ | 40.0 | 29.5 | 39.2 | **9.8** |
| $N = 30, ER = 2$ | 30.0 | 9.2 | 29.1 | **5.6** |
| $N = 50, ER = 2$ | 50.0 | 23.7 | 73.2 | **11.7** |
| $N = 80, ER = 2$ | 80.0 | 49.3 | 97.2 | **19.6** |

Table 5: **Results on Linear data on larger graphs.** Hamming distance $\mathcal{H}$ for learned and ground-truth edges on synthetic graphs, compared to other methods. The number of variables varies from 30 to 80, expected degree = 1 or 2. We compare to DCDI (Brouillard et al., 2020) and ENCO (Lippe et al., 2021) as they are the best performing baselines and they are able to handle larger graphs.

### A.5.2 RESULTS ON ANM DATA

For additive noise non-linear model (ANM) data, we compare to the strongest baseline models: DCDI and ENCO on $N \leq 80$ graphs. Results are found in Table 6. CSIvA achives hamming distance $\mathcal{H} < 11$ on all graphs of size up to 80. Therefore, CSIvA significantly outperforms strong baseline models DCDI and ENCO on all graphs. This difference becomes more apparent on larger graphs of $N \geq 50$.

| Dataset | All-absent | DCDI (Brouillard et al., 2020) | ENCO (Lippe et al., 2021) | CSIvA |
|---------|------------|-------------------------------|---------------------------|-------|
| $N = 30, ER = 1$ | 15.0 | 3.1 | 14.7 | **0.7** |
| $N = 50, ER = 1$ | 25.0 | 7.8 | 25.6 | **0.7** |
| $N = 80, ER = 1$ | 40.0 | 16.9 | 39.7 | **4.3** |
| $N = 30, ER = 2$ | 30.0 | 5.8 | 31.2 | **2.1** |
| $N = 50, ER = 2$ | 50.0 | 9.7 | 48.2 | **3.2** |
| $N = 80, ER = 2$ | 80.0 | 37.1 | 78.2 | **10.6** |

Table 6: **Results on continuous non-linear additive noise model (ANM) data with larger graphs.** Hamming distance $\mathcal{H}$ for learned and ground-truth edges on synthetic graphs, compared to other methods. The number of variables varies from 30 to 80, expected degree = 1 or 2. We compare to DCDI (Brouillard et al., 2020) and ENCO (Lippe et al., 2021) as they are the best performing baselines and they are able to handle larger graphs.

### A.5.3 RESULTS ON NN DATA

Results for comparisons between CSIvA and strongest baselines DCDI and ENCO on non-additive noise non-linear (NN) data are found in Table 7. CSIvA achieves hamming distance $\mathcal{H} < 11$ on all graphs. Hence, CSIvA significantly outperforms DCDI and ENCO on all graphs. The differences grow larger as the size of the graph grows.

Additionally, we generated non-additive noise non-linear (NN) data with scale-free (SF) graphs with degree of 4 and 6 for $N \leq 50$. We compare our model to strongest baselines DCDI and ENCO on these data. The results are found in Table 15.

| Dataset | All-absent | DCDI (Brouillard et al., 2020) | ENCO (Li et al., 2019) | CSIvA |
|---|---|---|---|---|
| $N = 30, ER = 1$ | 15.0 | 3.9 | 14.7 | **0.8** |
| $N = 50, ER = 1$ | 25.0 | 9.4 | 23.5 | **1.3** |
| $N = 80, ER = 1$ | 40.0 | 19.5 | 39.2 | **5.3** |
| $N = 30, ER = 2$ | 30.0 | 6.7 | 29.2 | **2.1** |
| $N = 50, ER = 2$ | 50.0 | 10.4 | 49.5 | **3.3** |
| $N = 80, ER = 2$ | 80.0 | 39.8 | 80.6 | **10.5** |

Table 7: **Results on continuous non-linear non-additive noise model (NN) data with larger Erdős–Rényi (ER) graphs.** Hamming distance $\mathcal{H}$ for learned and ground-truth edges on synthetic graphs, compared to other methods. The number of variables varies from 30 to 80, expected degree = 1 or 2. We compare to DCDI (Brouillard et al., 2020) and ENCO (Lippe et al., 2021) as they are the best performing baselines and they are able to handle larger graphs.

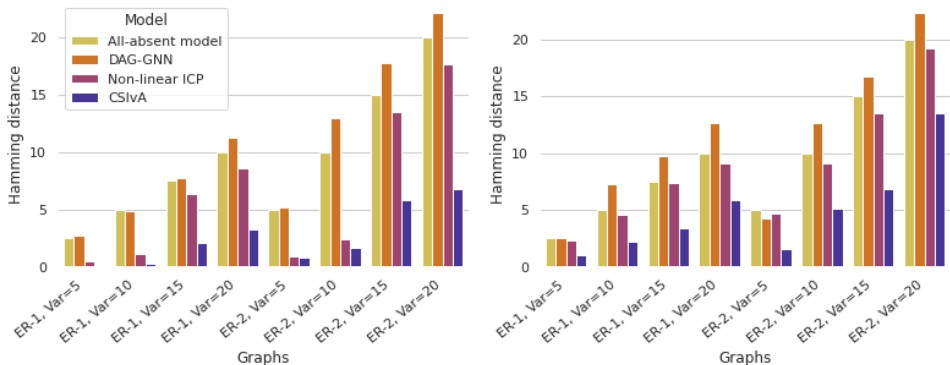

(a) **Results on Linear data on $N \leq 20$ graphs.**    (b) **Results on MLP data on $N \leq 20$ graphs.**

Figure 5: Hamming distance $\mathcal{H}$ between predicted and ground-truth adjacency matrices on the Linear and MLP data for $N \leq 20$ graphs, compared to DAG-GNN (Yu et al., 2019) and non-linear ICP (Heinze-Deml et al., 2018b), averaged over 128 sampled graphs. CSIvA significantly outperforms all other baselines.

### A.5.4    RESULTS ON MLP DATA

Results for comparisons between our model CSIvA and baselines non-linear ICP (Heinze-Deml et al., 2018b) and DAG-GNN (Yu et al., 2019) on MLP data are shown in Table 9. MLP data is non-linear and hence more challenging compared to the continuous linear data. Our model CSIvA significantly outperforms non-linear ICP and DAG-GNN. The difference becomes more apparent as the graph size grows larger and more dense.

| Dataset | All-absent | DCDI (Brouillard et al., 2020) | ENCO (Lippe et al., 2021) | CSIvA |
|---|---|---|---|---|
| $N = 30, ER = 1$ | 15.0 | 11.1 | 12.5 | **8.0** |
| $N = 50, ER = 1$ | 25.0 | 27.3 | 28.3 | **17.6** |
| $N = 80, ER = 1$ | 40.0 | 52.8 | 35.8 | **28.6** |
| $N = 30, ER = 2$ | 30.0 | 17.4 | 16.9 | **15.6** |
| $N = 50, ER = 2$ | 50.0 | 47.6 | 48.2 | **33.6** |
| $N = 80, ER = 2$ | 80.0 | 75.3 | 61.5 | **52.3** |

Table 8: **Results on discrete MLP data with larger graphs.** Hamming distance $\mathcal{H}$ for learned and ground-truth edges on synthetic graphs, compared to other methods. The number of variables varies from 30 to 80, expected degree = 1 or 2. We compare to DCDI (Brouillard et al., 2020) and ENCO (Lippe et al., 2021) as they are the best performing baselines and they are able to handle larger graphs.

|  | ER = 1 | | | | ER = 2 | | | |
|---|---|---|---|---|---|---|---|---|
|  | Var = 5 | Var = 10 | Var = 15 | Var = 20 | Var = 5 | Var = 10 | Var = 15 | Var = 20 |
| Abs. | 2.50 | 5.00 | 7.50 | 10.00 | 5.00 | 10.00 | 15.00 | 20.00 |
| **DAG-GNN** | 2.52 | 7.30 | 9.74 | 12.72 | 4.33 | 12.78 | 16.73 | 22.33 |
| **Non-linear ICP** | 2.43 | 4.62 | 7.42 | 9.05 | 4.76 | 9.12 | 13.52 | 19.25 |
| CSIvA | **0.98** $\pm$ 0.16 | **2.25** $\pm$ 0.17 | **3.38** $\pm$ 0.12 | **5.92** $\pm$ 0.19 | **1.51**$\pm$ 0.47 | **5.12** $\pm$ 0.26 | **6.82** $\pm$ 0.23 | **13.50** $\pm$ 0.35 |

Table 9: **Results on MLP data**. Hamming distance $\mathcal{H}$ for learned and ground-truth edges on synthetic graphs, compared to other methods, averaged over 128 sampled graphs ($\pm$ standard deviation). The number of variables varies from 5 to 20, expected degree = 1 or 2, and the dimensionality of the variables are fixed to 3. We compared to DAG-GNN (Yu et al., 2019), which is a strong baseline that uses observational data. We also compare to Non-linear ICP (Heinze-Deml et al., 2018b), which is a strong baseline that uses *interventional* data. Note that for (Heinze-Deml et al., 2018b), the method required nodes to be causally ordered, and Gaussian noise $\mathcal{N}(0, 0.1)$ to be added. "Abs" baselines are All-Absent baselines, which is an baseline model that outputs all zero edges for the adjacency matrix.

### A.5.5 RESULTS ON DIRICHLET DATA.

We run two sets of experiments on Dirichlet data. The first set of experiments is aimed at understanding how different $\alpha$ values impact the performance of our model. In the second set of experiments, we compare the performance of our model to four strong baselines: DCDI (Brouillard et al., 2020), ENCO (Lippe et al., 2021), non-linear ICP (Heinze-Deml et al., 2018b) and DAG-GNN (Yu et al., 2019).

|  | ER = 1 | | | | ER = 2 | | | |
|---|---|---|---|---|---|---|---|---|
|  | Var = 5 | Var = 10 | Var = 15 | Var = 20 | Var = 5 | Var = 10 | Var = 15 | Var = 20 |
| $\alpha = 0.1$ | 0.18$\pm$ 0.03 | 0.72$\pm$ 0.04 | 1.31$\pm$ 0.04 | 2.45 $\pm$ 0.04 | 0.39$\pm$ 0.04 | 1.27$\pm$ 0.07 | 1.98$\pm$ 0.12 | 4.09$\pm$ 0.04 |
| $\alpha = 0.25$ | 0.14$\pm$ 0.03 | 0.77$\pm$ 0.05 | 1.62$\pm$ 0.05 | 3.51 $\pm$ 0.05 | 0.29$\pm$ 0.04 | 1.27$\pm$ 0.07 | 3.04$\pm$ 0.20 | 6.41$\pm$ 0.12 |
| $\alpha = 0.5$ | 0.14$\pm$ 0.04 | 0.94$\pm$ 0.05 | 4.26$\pm$ 0.07 | 7.35$\pm$ 0.04 | 0.41 $\pm$ 0.03 | 2.11$\pm$ 0.06 | 8.25$\pm$ 0.07 | 15.54$\pm$ 0.10 |
| $\alpha = 1.0$ | 0.26$\pm$ 0.05 | 2.37$\pm$ 0.07 | 4.90$\pm$ 0.05 | 10.10$\pm$ 0.07 | 0.68$\pm$ 0.03 | 4.32$\pm$ 0.07 | 10.24$\pm$ 0.07 | 21.81$\pm$ 0.07 |
| $\alpha = 5.0$ | 1.27$\pm$ 0.12 | 4.9$\pm$ 0.05 | 14.73$\pm$ 0.11 | 19.49$\pm$ 0.05 | 3.21$\pm$ 0.05 | 9.99 $\pm$ 0.03 | 24.19$\pm$ 0.05 | 37.03$\pm$ 0.24 |
| Abs. | 2.5 | 5.0 | 7.5 | 10.0 | 5.0 | 10.0 | 15.0 | 20.0 |

Table 10: **Results on Dirichlet data.** Hamming distance $\mathcal{H}$ (lower is better) for learned and ground-truth edges on synthetic graphs, averaged over 128 sampled graphs. Our model accomplished a hamming distance of less than 2.5 for Dirichlet data with $\alpha \leq 0.5$. "Abs" baselines are All-Absent baselines, which is an baseline model that outputs all zero edges for the adjacency matrix.

The results for the first set of experiments are shown in Table 10. Our model performs well on all graphs where $\alpha \leq 0.5$, and the performance starts to degrading as $\alpha = 1.0$. When $\alpha = 5.0$, our model is almost performing similarly to the All-absent model (outputting all zero edges). This is to be expected, as larger alpha values is less informative of the causal relationships between variables.

For the second set of experiments, we compared CSIvA to non-linear ICP and DAG-GNN on graphs of size up to 20. To limit the number of experiments to run, we set $\alpha = 1.0$, as this gives the least amount of prior information to CSIvA. As shown in Figure 6, our model significantly outperforms non-linear ICP and DAG-GNN. Our model achieves $\mathcal{H} < 5$ on size 10 graphs, almost half of the error rate compared

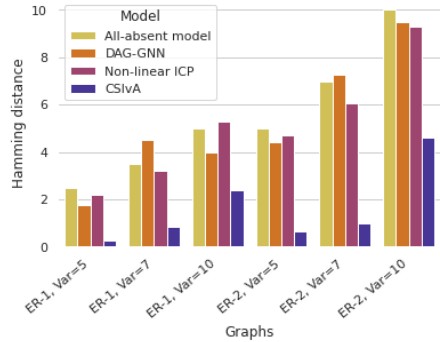

Figure 6: **Results on Dirichlet data on $N \leq 20$ graphs**. Hamming distance $\mathcal{H}$ between predicted and ground-truth adjacency matrices on Dirichlet data, averaged over 128 sampled graphs.

| | ER = 1 | | | ER = 2 | | |
|---|---|---|---|---|---|---|
| | **Var = 5** | **Var = 7** | **Var = 10** | **Var = 5** | **Var = 7** | **Var = 10** |
| All-absent baseline | 2.5 | 3.5 | 5.0 | 5.0 | 7.0 | 10.0 |
| (Yu et al., 2019) | 1.75 | 4.5 | 4.0 | 4.5 | 7.25 | 9.50 |
| (Heinze-Deml et al., 2018b) | 2.2 | 3.2 | 5.3 | 4.7 | 6.1 | 9.3 |
| CSIvA | **0.26** $_{\pm\,0.05}$ | **0.83** $_{\pm\,0.06}$ | **2.37** $_{\pm\,0.07}$ | **0.65** $_{\pm\,0.05}$ | **0.97** $_{\pm\,0.06}$ | **4.59** $_{\pm\,0.08}$ |

Table 11: **Results on Dirichlet data.** Hamming distance $\mathcal{H}$ for learned and ground-truth edges on synthetic graphs, compared to other methods, averaged over 128 sampled graphs ($\pm$ standard deviation). The number of variables varies from 5 to 10, expected degree = 1 or 2, the dimensionality of the variables are fixed to 3, and the $\alpha$ is fixed to 1.0. We compare to the strongest causal-induction methods that uses observational data (Yu et al., 2019) and the strongest that uses *interventional* data (Heinze-Deml et al., 2018b).

to non-linear ICP and DAG-GNN, both achieving a significantly higher Hamming distance ($\mathcal{H} = 9.3$ and $\mathcal{H} = 9.5$ respectively) on larger and denser graphs. Refer to Table 11 for complete sets of results.

For larger graphs ($N > 20$), we compare CSIvA to DCDI (Brouillard et al., 2020) and ENCO (Lippe et al., 2021) as they could scale to larger graphs and are the strongest baselines. The results are illustrated in Figure 3 and the detailed results are found in Table 12. As the graphs get larger, the performance of baselines DCDI and ENCO drops significantly, for dense and largr graphs (ER-2, $N = 80$), the baseline models achieve almost near all-absent performance, while our model performs significantly better (achieving almost $30\%$ performance gain in terms of structured hamming distance).

| Dataset | All-absent | DCDI (Brouillard et al., 2020) | ENCO (Lippe et al., 2021) | CSIvA |
|---|---|---|---|---|
| $N = 30, ER = 1$ | 15.0 | 8.3 | 4.9 | **2.9** |
| $N = 40, ER = 1$ | 20.0 | 16.2 | 9.9 | **5.3** |
| $N = 50, ER = 1$ | 25.0 | 26.4 | 17.7 | **8.9** |
| $N = 60, ER = 1$ | 30.0 | 28.4 | 25.6 | **12.6** |
| $N = 70, ER = 1$ | 35.0 | 34.1 | 29.1 | **18.4** |
| $N = 80, ER = 1$ | 40.0 | 54.1 | 39.1 | **23.1** |
| $N = 30, ER = 2$ | 30.0 | 12.3 | 8.1 | **6.5** |
| $N = 40, ER = 2$ | 40.0 | 21.6 | 14.3 | **9.6** |
| $N = 50, ER = 2$ | 50.0 | 47.6 | 28.6 | **17.3** |
| $N = 60, ER = 2$ | 60.0 | 57.6 | 38.2 | **26.6** |
| $N = 70, ER = 2$ | 70.0 | 71.2 | 53.4 | **40.9** |
| $N = 80, ER = 2$ | 80.0 | 78.6 | 72.6 | **56.3** |

Table 12: **Results on Dirichlet data with larger graphs.** Hamming distance $\mathcal{H}$ for learned and ground-truth edges on synthetic graphs, compared to other methods. The number of variables varies from 30 to 80, expected degree = 1 or 2, and the $\alpha$ is fixed to 0.1. We compare to the approaches from (Brouillard et al., 2020) and (Lippe et al., 2021) as they are the strongest baselines and they are able to handle larger graphs.

### A.5.6 RESULTS ON SCALE-FREE GRAPHS

We further evaluated our model's performance on on scale-free (SF) graphs. We run experiments for both discrete and continuous variables. For discrete variables. We train and test our model on Dirichlet data with $\alpha = 0.1$ on scale-free (SF) graphs. We run them on SF graphs with degree 4 and 6 similar to the setup in Zheng et al. (2018). Results on Dirichlet data is reported in Table 14 and results on NN data is found in Table 15. Our model significantly outperforms strong baseline models DCDI (Brouillard et al., 2020) and ENCO (Lippe et al., 2021) in all cases, regardless of the type of data, size of the graph and the sparsity of the graphs.

|  | ER = 1 | | | ER = 2 | | |
|---|---|---|---|---|---|---|
|  | Var = 10 | Var = 15 | Var = 20 | Var = 10 | Var = 15 | Var = 20 |
| All-absent baseline | 5.0 | 7.5 | 10.0 | 10.0 | 15.0 | 20.0 |
| $\alpha = 0.1$ | 0.5 | 1.2 | 2.4 | 0.9 | 2.3 | 3.4 |
| $\alpha = 0.25$ | 0.5 | 1.7 | 3.3 | 1.7 | 3.5 | 4.6 |

Table 13: **Results on imperfect interventions.** Hamming distance $\mathcal{H}$ for learned and ground-truth edges on synthetic graphs, compared to other methods, averaged over 128 sampled graphs ($\pm$ standard deviation). The number of variables varies from 10 to 20, expected degree = 1 or 2, the dimensionality of the variables are fixed to 3, and the $\alpha$ is fixed to 0.1 or 0.25.

| Dataset | All-absent | DCDI(Brouillard et al., 2020) | ENCO(Lippe et al., 2021) | CSIvA |
|---|---|---|---|---|
| $N = 10, SF = 4$ | 20.0 | 5.1 | 3.1 | **0.7** |
| $N = 10, SF = 6$ | 30.0 | 7.6 | 5.2 | **1.7** |
| $N = 20, SF = 4$ | 40.0 | 11.7 | 8.1 | **3.2** |
| $N = 20, SF = 6$ | 60.0 | 13.1 | 9.3 | **3.7** |
| $N = 30, SF = 4$ | 60.0 | 17.4 | 12.7 | **4.9** |
| $N = 30, SF = 6$ | 90.0 | 21.2 | 17.1 | **9.1** |
| $N = 50, SF = 4$ | 100.0 | 68.1 | 43.7 | **25.8** |
| $N = 50, SF = 6$ | 150.0 | 70.6 | 47.1 | **30.3** |

Table 14: **Results on Dirichlet data on Scale-Free graphs.** Hamming distance $\mathcal{H}$ for learned and ground-truth edges on synthetic graphs, compared to other methods. The number of variables varies from 30 to 80, expected degree = 4 or 6, and the $\alpha$ is fixed to 0.1. We compare to the approaches from Brouillard et al. (2020) and Lippe et al. (2021) as they are the strongest baselines and they are able to handle larger graphs.

### A.5.7 Soft interventions

In addition to hard interventions, we also consider soft interventions. Hence, we further evaluate our model on imperfect interventions as discussed in Section 6.1. To limit the number of experiments to run, we focus on Dirichlet data with $\alpha = 0.1$ and $\alpha = 0.25$.

Soft interventions on Dirichlet data is generated as follows. For an intervention on variable $X_i$, we first sample the $\alpha_i$ for the Dirichlet distribution from a Uniform distribution $U\{0.1, 0.3, 0.5, 0.7, 0.9\}$, then using the sampled $\alpha_i$, we sample the conditional probabilities from the Dirichlet distribution with the new $\alpha_i$.

The results for CSIvA on soft interventions are shown in Table 13. CSIvA was able to achieve a hamming distance $\mathcal{H} < 5$ on all graphs of size $H \leq 20$. These results are strong indications that that our model still works well on the imperfect interventions.

### A.5.8 Unknown Interventions

Until now, we have considered cases where the target of the intervention is known, we now consider the case when the target of the intervention is unknown, this is also referred to as unknown interventions. Again, to limit the number of experiments to run, we focus on Dirichlet data with $\alpha = 0.1$ and $\alpha = 0.25$. The model does not know the target of the intervention and all other training procedures remains exactly the same. The results are shown in Table 16. We compare how well CSIvAperforms on known and unknown interventions. We can see that the performance for the known and unknown interventions are almost the same for sparser graphs (ER-1). The differences increases slightly for denser graphs with higher values of $\alpha$, for example $ER - 2$ graphs with $\alpha = 0.25$. This is to be expected, as denser graphs with higher $\alpha$ values are more challenging to learn. The biggest differences for the performances of known and unknown interventions is less than $\mathcal{H} < 1.0$ in hamming distance. These results shown clear indications that our model performs well even when the intervention target is unknown.

| Dataset | All-absent | DCDI(Brouillard et al., 2020) | ENCO(Lippe et al., 2021) | CSIvA |
|---|---|---|---|---|
| $N = 10, SF = 4$ | 20.0 | 4.7 | 2.7 | **0.9** |
| $N = 10, SF = 6$ | 30.0 | 6.8 | 4.4 | **1.0** |
| $N = 20, SF = 4$ | 40.0 | 9.2 | 6.5 | **1.5** |
| $N = 20, SF = 6$ | 60.0 | 13.1 | 8.7 | **3.4** |
| $N = 30, SF = 4$ | 60.0 | 13.7 | 9.1 | **1.8** |
| $N = 30, SF = 6$ | 90.0 | 19.4 | 14.7 | **6.4** |
| $N = 50, SF = 4$ | 100.0 | 25.5 | 19.1 | **6.9** |
| $N = 50, SF = 6$ | 150.0 | 40.1 | 29.3 | **19.2** |

Table 15: **Results on continuous non-linear non-additive noise model (NN) data on Scale-Free (SF) graphs.** Hamming distance $\mathcal{H}$ for learned and ground-truth edges on synthetic graphs, compared to other methods. The number of variables varies from 30 to 80, expected degree = 4 or 6, and the $\alpha$ is fixed to $0.1$. We compare to the approaches from Brouillard et al. (2020) and Lippe et al. (2021) as they are the strongest baselines and they are able to handle larger graphs.

| | ER = 1 | | | | ER = 2 | | | |
|---|---|---|---|---|---|---|---|---|
| | Var = 15 | | Var = 20 | | Var = 15 | | Var = 20 | |
| | $\alpha = 0.1$ | $\alpha = 0.25$ | $\alpha = 0.1$ | $\alpha = 0.25$ | $\alpha = 0.1$ | $\alpha = 0.25$ | $\alpha = 0.1$ | $\alpha = 0.25$ |
| Known | 1.3 | 1.7 | 2.5 | 3.4 | 2.3 | 3.0 | 4.1 | 6.3 |
| Unknown | 1.4 | 1.9 | 2.5 | 3.3 | 2.3 | 3.5 | 4.1 | 6.6 |

Table 16: **Results on unknown interventions compared to known interventions.** Hamming distance $\mathcal{H}$ from learned and ground-truth edges on synthetic graphs, for known vs unknown interventions, averaged over 128 sampled graphs ($\pm$ standard deviation). The number of variables varies from 15 to 20, expected degree = 1 or 2, $\alpha = 0.1$ or $0.25$, and the dimensionality of the variables are fixed to 3.

## A.6 DETAILED RESULTS FOR OUT-OF-DISTRIBUTION EXPERIMENTS

This section contains detailed results for the out-of-distribution experiments experiments in Section 6.2.

### A.6.1 VARYING GRAPH DENSITY

Results for the experiments on varying graph density in Section 6.2 are shown in Table 17 and Table 18. Our model generalizes well to OOD test distribution where the graphs vary in terms of density.

### A.6.2 VARYING CONDITIONAL DISTRIBUTION

The results on varying $\alpha$ values in Section 6.2 are found in Table 19. Our model generalizes well to OOD test distributions, where the conditional probability can vary in terms of $\alpha$ values for the Dirichlet distributions.

| **Train** | | ER-1 | ER-2 | ER-3 |
|---|---|---|---|---|
| | ER-1 | 1.2 | 0.9 | 1.3 |
| **Test** | ER-2 | 3.3 | 1.8 | 2.1 |
| | ER-3 | 5.0 | 2.8 | 2.8 |

Table 17: **Results on varying graph density for MLP data:** Hamming distance $\mathcal{H}$ between predicted and ground-truth adjacency matrices.

| **Train** | | ER-1 | ER-2 | ER-3 |
|---|---|---|---|---|
| | ER-1 | 0.19 | 0.21 | 0.28 |
| **Test** | | 0.86 | 0.29 | 0.25 |
| | ER-3 | 1.61 | 0.60 | 0.23 |

Table 18: **Results on graph sparsity for Dirichlet data** ($\alpha = 1$)**:** Hamming distance $\mathcal{H}$ between predicted and ground-truth adjacency matrices.

| Train | | $\alpha = 0.1$ | $\alpha = 0.25$ | $\alpha = 0.5$ |
|---|---|---|---|---|
| | $\alpha = 0.1$ | 0.31 | 0.33 | 0.52 |
| **Test** | $\alpha = 0.25$ | 0.72 | 0.40 | 0.41 |
| | $\alpha = 0.5$ | 1.8 | 0.71 | 0.35 |

Table 19: **Results on varying $\alpha$ values for Dirichlet data:** Hamming distance $\mathcal{H}$ between predicted and ground-truth adjacency matrices.

| Train | N=10 | ER-2 | ER-3 | ER-4 |
|---|---|---|---|---|
| | SF-2 | 1.2 | 0.9 | 1.3 |
| **Test** | SF-3 | 3.3 | 1.8 | 2.1 |
| | SF-4 | 5.0 | 2.8 | 2.8 |

Table 20: **Results on varying graph models for Dirichlet data for $N = 10$ graphs:** Hamming distance $\mathcal{H}$ between predicted and ground-truth adjacency matrices for Dirichlet data trained on Erdős–Rényi (ER) graphs and tested on scale-free (SF) graphs.

| Train | N=20 | ER-2 | ER-3 | ER-4 |
|---|---|---|---|---|
| | SF-2 | 0.19 | 0.21 | 0.28 |
| **Test** | SF-3 | 0.86 | 0.29 | 0.25 |
| | SF-4 | 1.61 | 0.60 | 0.23 |

Table 21: **Results on varying graph models for Dirichlet data for $N = 20$ graphs:** Hamming distance $\mathcal{H}$ between predicted and ground-truth adjacency matrices for Dirichlet data trained on Erdős–Rényi (ER) graphs and tested on scale-free (SF) graphs.

### A.6.3 VARYING GRAPH MODELS

We also evaluate how well our model generalizes when the model used for generating the graphs during training and test differs. We trained and tested our model on Erdős–Rényi (ER) and Scale-Free (SF) graph models. To limit the number of experiments to run, we evaluate our model on Dirichlet data with $\alpha = 0.1$. We run experiments both on $N = 10$ and $N = 20$ graphs. The models are trained on Erdős–Rényi (ER) graphs and tested on Scale-Free (SF) graphs. The results are shown in Table 20 and Table 21.

### A.7 ABLATION STUDIES AND ANALYSES

In order to better understand the performance of our model, we performed further analysis and ablations. This section contains full results for Section 6.4. In this section, we aim to answer the following questions:

- What does visualizations of the generated graphs look like?
- Additional evaluation metrics to measure structural accuracy.
- Are these generated graphs from our model acyclic?
- Does intervention data improve identifiability in our model?
- How does varying the number of samples impact the performance of our model?
- How does varying the amount of training datasets impact the peformance of our model?
- How does different components of our model (such as sample-level attention and auxiliary loss) impact the performance of our model?

### A.7.1 VISUALIZATION OF SAMPLED GRAPHS

We visualized samples that our model generated on the test data. The samples are shown in Figure 8 and Figure 7. The samples are randomly chosen, each subplot is a sample from a distinct test data. The edges in the graph are shown in 3 colors, they each represent the following: (a) Green edges indicate that our model has generated the correct edge. (b) A red edge indicates a missing edge, that is our model did not generate the edge, which exist in the groundtruth graph. (c) A blue edge indicates

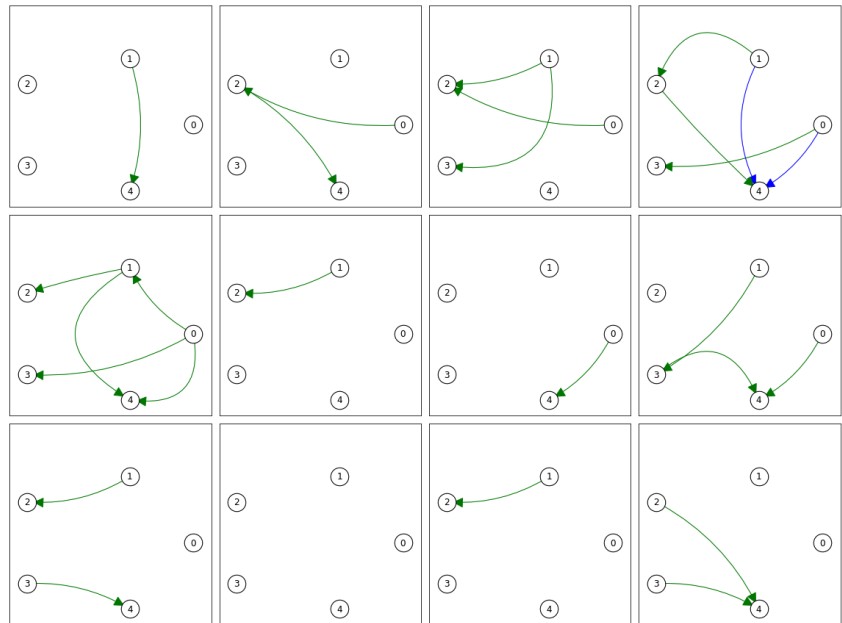

Figure 7: This figures visualizes samples that our model generated on test data. The model was trained and tested on MLP data of size 5 with **ER-**1 graphs. The samples are randomly chosen. The green edges indicate that our model has generated the correct edges; red edges indicate edges that our model had missed; and blue edges are the ones that our model generated, which were not in the groundtruth graph. As shown above, our model is able to generate the correct graph almost all of the times, while only occasionally generating 1 or 2 incorrect edges in a graph.

a redundant edge, such that our model generated an edge that does not exist in the groundtruth graph. As shown in Figure 8 and 7, our model is able to generate the correct graph almost all of the times.

### A.7.2 ADDITIONAL EVALUATIONS

We evaluated our model on additional evaluation metrics to better understand the performance of the model. In particular, we evaluated our model on the balanced scoring function (BSF), F1 score, precision, recall and area under roc (AUROC). We evaluated our model both on discrete and continous data. For discrete data, we used Dirichlet data with $\alpha = 0.1$ and for continous data, we used non-linear non-additive noise (NN) data. Results on Dirichlet data is in Table 22, results on NN data is in Table 23.

For Dirichlet data, our model achieved strong results. Again, the area under ROC (AUROC) is greater than 0.95 for all cases. The F1 score is also always above 0.90. For continuous data, our model achieved even stronger results. Our model scored between 0.98 and 0.99 for all scores. This is an indication that the edges were almost always correct and the correct edges also have strong probability.

| Number of variables | Graph sparsity | SHD | BSF | F1 score | Precision | Recall | AUROC |
|---|---|---|---|---|---|---|---|
| $N = 10$ | $ER = 1$ | 0.5 | 0.91 | 0.95 | 0.97 | 0.93 | 0.96 |
| $N = 10$ | $ER = 2$ | 1.0 | 0.93 | 0.95 | 0.96 | 0.93 | 0.98 |
| $N = 20$ | $ER = 1$ | 2.4 | 0.91 | 0.90 | 0.88 | 0.92 | 0.99 |
| $N = 20$ | $ER = 2$ | 3.3 | 0.94 | 0.94 | 0.91 | 0.95 | 0.99 |

Table 22: **Results on Dirichlet data evaluated with additional evaluation metrics** Hamming distance $\mathcal{H}$ for learned and ground-truth edges on synthetic graphs, compared to other methods. The number of variables varies from 10 to 20, expected degree = 1 or 2, and the $\alpha$ is fixed to $0.1$.

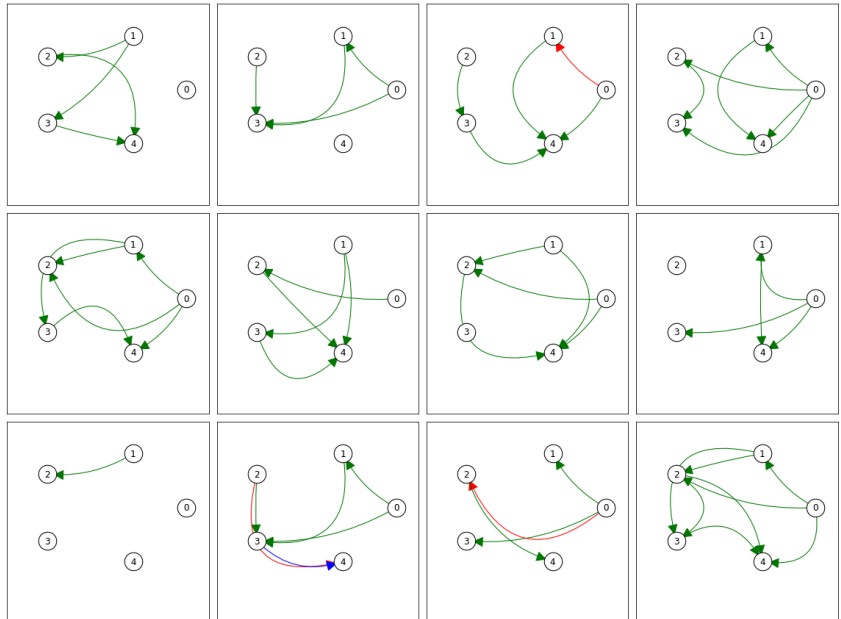

Figure 8: This figures visualizes samples that our model generated on test data. The model was trained and tested on MLP data of size 5 with **ER-**2 graphs. The samples are randomly chosen. The green edges indicate that our model has generated the correct edges; red edges indicate edges that our model had missed; and blue edges are the ones that our model generated, which were not in the groundtruth graph. As shown above, our model is able to generate the correct graph almost all of the times, while only occasionally generating 1 or 2 incorrect edges in a graph.

| Number of variables | Graph sparsity | SHD | BSF | F1 score | Precision | Recall | AUROC |
|---|---|---|---|---|---|---|---|
| $N = 10$ | $ER = 1$ | 0.17 | 0.98 | 0.99 | 0.99 | 0.98 | 0.93 |
| $N = 10$ | $ER = 2$ | 0.30 | 0.98 | 0.99 | 0.99 | 0.98 | 0.93 |
| $N = 20$ | $ER = 1$ | 0.12 | 0.99 | 0.99 | 0.99 | 0.99 | 0.92 |
| $N = 20$ | $ER = 2$ | 0.62 | 0.99 | 0.99 | 0.99 | 0.99 | 0.95 |

Table 23: **Results on NN data evaluated with additional evaluation metrics** Hamming distance $\mathcal{H}$ for learned and ground-truth edges on synthetic graphs, compared to other methods. The number of variables varies from 10 to 20, expected degree = 1 or 2.

### A.7.3  ACYCLICITY OF GENERATED GRAPHS

As discussed in Section 6.4, we analyzed the generated graphs for acyclicity on Dirichlet data with $\alpha = 0.1$ and $\alpha = 0.25$ for graphs of size up to $N \leq 20$. We evaluated the model on 128 test datasets per setting, and found that none of the generated graphs contains cycles, all graphs are acyclic. This is a clear indication showing that although we do not have an acyclicity regularizer, our model was still able to learn to generated DAGs.

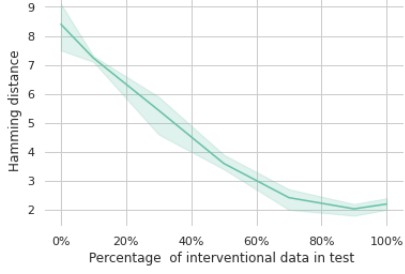

### A.7.4  IDENTIFIABILITY
UPON SEEING INTERVENTION DATA.

Our previous experiments were all performed using a fixed amount of interventions (80%) in

Figure 9: **Results on varying amount of interventions in data.** Hamming distance $\mathcal{H}$ for learned and ground-truth edges on synthetic Dirichlet graphs ($N = 15$ and $\alpha = 0.25$). Pure observational data (0% interventions) performs the worst, which is to be expect, since intervention data is need for causal identifiability. The performance of our model improves as it observes more interventions, this suggest that our model is able to extract useful information from interventions in order to predict the causal structure.

the training and test sets. To investigate how changing the proportion of interventional data impacts the performance of our model, we train the model with varying amounts of interventions in the training set and evaluate it using different amount of interventions during test time.

To be specific, during training, the model is trained on data with varying amount of interventions, which is randomly sampled from the uniform distribution $U[0, 0.1, 0.3, 0.5, 0.7, 0.9, 1.0]$. During test time, we evaluate the model on different amount of interventions, and we report the performance in hamming distance. We trained the model on Dirichlet data with 15 nodes and $\alpha = 0.25$. The model is trained and tested on 1000 samples per dateset. The results are found in Figure 9.

As shown in Figure 9, our model's performance is worst if it only receives observational data (0% interventions), and the performance of our model improves as the amount of interventional data increases. This is a clear indication that our model is able to extract information from interventional data for predicting the graph structure.

### A.7.5 Varying number of samples

We evaluated CSIvA on different amount of samples $(100, 200, 500, 1000, 1500)$ per CBNs. To limit the number of experiments to run, we focus on Dirichlet data sampled from $N = 10$ graphs. During the $ith$ iteration in training, the model takes in as input a dataset with $l_i$ samples, where $l_i$ is sampled from $U[100, 200, 500, 1000, 1500]$. During test time, we run separate evaluations on the model for test datasets with different number of samples and the results are reported in Figure 10. We can see that the model performance improves as it observes up to 1000 samples for ER-1 graphs, whereas having 1500 samples gives slightly better results compared to 1000 samples for ER-2 graphs.

Figure 10: **Results on varying number of samples**. Hamming distance $\mathcal{H}$ between predicted and ground-truth adjacency matrices for synthetic data. Results for CSIvA trained on Dirichlet data with $N = 10$ and $\alpha = 0.5$ with different numbers of samples per CBNs. The model performance increases as the sample size increases.

### A.7.6 Varying amount of training data

In this section, we evaluate how the amount of training data impacts the performance of our model (CSIvA). To limit the number of experiments to run, we focus on Dirichlet data with $\alpha = 0.1$. The model receives $M$ amount of training datasets, where $M$ is sampled from $U[5000, 10000, 15000, 20000]$.

| Dataset | $I = 5000$ | $I = 10000$ | $I = 15000$ | $I = 20000$ |
|---|---|---|---|---|
| $N = 10, ER = 1$ | 1.5 | 1.1 | 1.0 | **0.8** |
| $N = 10, ER = 2$ | 3.2 | 2.2 | 1.7 | **1.4** |
| $N = 25, ER = 1$ | 6.9 | 2.8 | 1.7 | **1.4** |
| $N = 25, ER = 2$ | 13.5 | 7.2 | 5.8 | **5.2** |
| $N = 50, ER = 1$ | 32.0 | 12.2 | 11.3 | **8.9** |
| $N = 50, ER = 2$ | 42.0 | 30.3 | 23.3 | **17.3** |

Table 24: **Results on Dirichlet data trained with varying amount of training datasets.** Hamming distance $\mathcal{H}$ for learned and ground-truth edges on synthetic graphs, compared to other methods. The number of variables varies from 10 to 50, expected degree = 1 or 2, and the $\alpha$ is fixed to 0.1.

### A.7.7 Interventions on a few variables

We performed additional experiments to understand the performance of our model when only a few variables are intervened on. We varied the percentage of variables to intervene on from 10% to 100%.

Results are reported in Table 25. We used Dirichlet data with $alpha = 0.1$ for both training and test data. Our model's performance improves consistently as more nodes are intervened on.

| Dataset | 10% | 30% | 50% | 70% | 90% | 100% |
|---|---|---|---|---|---|---|
| $N = 10, ER = 1$ | 6.4 | 5.0 | 3.2 | 2.2 | 0.9 | 0.4 |
| $N = 10, ER = 2$ | 11.9 | 9.5 | 7.6 | 4.8 | 3.0 | 1.7 |

Table 25: **Results on Dirichlet data trained with varying percentage of intervention nodes.** Hamming distance $\mathcal{H}$ for learned and ground-truth edges on synthetic graphs, compared to other methods. The percentage of node that the model can intervene on varies from 10% 100%. The performance of the model improves consistently as it more nodes are intervened on.

### A.7.8 ABLATION STUDIES

We investigate the role that different components play in our model, for example, what are the effects of having auxillary loss and sample-level attention for our model. The details for each experiment is described below.

**Sample-level attention**    We first evaluated the performance of our with and without sample-level attention. In the case without sample-level attention, our model will only use node-level attention for all 10 layers in the encoder transformer. The results are reported in Table 26. The models with sample-attention is a vanilla CSIvA model; and the model without sample-attention is CSIvA without sample-level attention. As one could see that the performance of the model drops significantly if sample-attention is not used. Hereby making sample-level attention a crucial component in our model.

| | ER = 1 | | | | ER = 2 | | | |
|---|---|---|---|---|---|---|---|---|
| | Var = 15 | | Var = 20 | | Var = 15 | | Var = 20 | |
| | $\alpha = 0.1$ | $\alpha = 0.25$ | $\alpha = 0.1$ | $\alpha = 0.25$ | $\alpha = 0.1$ | $\alpha = 0.25$ | $\alpha = 0.1$ | $\alpha = 0.25$ |
| Sample attention | 1.3 | 1.7 | 2.5 | 3.4 | 2.3 | 3.0 | 4.1 | 6.3 |
| No sample attention | 15.3 | 15.5 | 21.8 | 22.4 | 29.3 | 29.7 | 39.4 | 40.0 |

Table 26: Results for CSIvA with and without **sample-level attention**. Hamming distance $\mathcal{H}$ from learned and ground-truth edges on synthetic graphs, for known vs unknown interventions, averaged over 128 sampled graphs ($\pm$ standard deviation). The number of variables varies from 15 to 20, expected degree = 1 or 2, $\alpha = 0.1$ or 0.25, and the dimensionality of the variables are fixed to 3.

**Auxiliary loss**    We also conducted ablation studies to understand the effect of the auxiliary loss in the objective function, results are reported in Table 27. The model with auxiliary loss is the vanilla CSIvA model, where as the one without is CSIvA trained without the auxiliary loss objective. Experiments are conducted on Dirichlet data with $\alpha = 0.1$, the model does gain a small amount of performance ($\mathcal{H} < 1.0$) by having the auxiliary loss, the difference becomes more apparent as the size of the graph grows, indicating that auxiliary loss plays an important role as a part of the objective function.

### A.7.9 COMPUTATION TIME

All the models with the proposed method are trained for $500k$ iterations and all models have converged by then (in fact they have all converged by $400k$). Once the model is trained, evaluation/ inference is fast. All models are evaluated on 128 test datasets (graphs).

The focus of causal discovery is on high-stakes test cases where accuracy matters more than compute. All models (baselines and our model) are trained on GPUs. For baseline methods like DCDI, each test

| | ER = 1 | | ER = 2 | |
| --- | --- | --- | --- | --- |
| | Var = 15 | Var = 20 | Var = 15 | Var = 20 |
| Aux loss | 1.2 | 2.4 | 2.3 | 3.4 |
| No aux loss | 1.8 | 3.1 | 2.8 | 4.5 |

Table 27: Results for CSIvA trained with and without **auxiliary loss**. Hamming distance $\mathcal{H}$ from learned and ground-truth edges on synthetic graphs, for known vs unknown interventions, averaged over 128 sampled graphs ($\pm$ standard deviation). The number of variables varies from 15 to 20, expected degree = 1 or 2, $\alpha = 0.1$, and the dimensionality of the variables are fixed to 3.

case can take up to $48$ hours to compute; baseline model ENCO takes up to 30 minutes to compute. In contrast, our method can take up to $8$ hours to train for large graphs, and then inference takes minutes. (More details are provided below.) The training time of our model can be amortized if the model can be used on multiple test cases; in contrast, training time for ENCO and DCDI is proportional to the number of test cases.

For our model CSIvA , training on smaller graphs ($N \leq 20$) converges within $50k$, which takes up to 45 minutes. Medium sized graphs ($20 < N \leq 50$) converge within $250k$ and take 4 hours. Large graphs ($50 < N < 80$) converge within $400k$ and take 8 hours. Note that once the model is trained, it is very fast to evaluate on different test cases. Our model is evaluated on $128$ test cases.

For the baseline model DCDI takes up to 4 hours on a single graph of size up to 20 ($N < 20$). The model takes up to 12 hours on a medium size graph ($N < 50$) and 48 hours for larger graphs ($50 < N < 80$). For the baseline model ENCO. Smaller graphs ($N < 20$) take 20 minutes per graph and larger graphs ($N < 80$) take 30 minutes per graph. To compute 128 test cases, ENCO takes 2.5 days of computation time.

