# OpenReview forum: "Learning to Induce Causal Structure "
_ICLR.cc/2023/Conference — ICLR 2023 poster_

### Official Review · Reviewer_s1nT · 2022-10-19

**Confidence:** 3
**Correctness:** 4
**Technical Novelty And Significance:** 3
**Empirical Novelty And Significance:** 3
**Recommendation:** 8

**Clarity, Quality, Novelty And Reproducibility:**

The work is fairly clear, though as a non-expert in neural networks some aspects seemed slightly vague to me.  Again, as a non-expert the work seems novel.  Reproducibility is harder to gauge, since no code is provided to replicate the results.


**Strength And Weaknesses:**

UPDATE: The reviewers have answered my concerns, so I'm upping my score to 8.

----

The empirical results look impressive, and it seems to beat all existing methods at this task.

The main weakness is that there is no discussion of how much training is required for CSIvA to obtain the results it does.  How much synthetic data must be generated, and how long does the computation take?  How does this compare to the other methods you consider?

Another weakness is that the method does not enforce acyclicity of the resulting graph, though the authors report that empirically this is never a problem.


**Summary Of The Paper:**

This paper provides an algorithm for learning the structure of a directed causal model from a combination of observational and interventional data, where interventions consist of setting one node to a specific value.


**Summary Of The Review:**

UPDATE: My score has been increased to 8.

----

The simulations are thorough and the results do seem promising, but I would have to hear a convincing answer to the question about runtimes, and have a link to the authors' code before I were to revise my score upwards.

### Minor Comments

- Figure 4 doesn't seem well presented.  It would seem more sensible (to me) to group the graphs by ER-1 vs ER-2, and then have separate line plots over the increasing number of nodes.  This would make it easier to visualize the best methods.
- Table 1 seems limited in terms of the methods listed.  What about the others you use in your comparisons?  (i.e. unsupervised approaches)
- page 8: "BnLearn" should be "bnlearn"
- Your spelling is inconsistent.  For example, on page 3 in the top paragraph you use both "modeling" and "modelling", and then on page 6 you use both "modelling" (not North America) and "analyzing" (only North America).
- The capitalization of "Hamming" is also not consistent, even within A.5.  I would suggest always capitalizing, as it is someone's name.
- Please ensure that the capitalization of your references is correct (use {} around such words in article titles in BibTeX).  e.g. "dag", "markov", "bayesian", "kaggle", "Chalearn", "Innovations in machine learning" etc.
- Caption for Table 11: "<=" should be "\leq", and the opening quote on 'Abs' is not inverted.
- Tables 5-8 and 12 - again, it would seem better to group the ER=1 and ER=2 results together.

---

> ### Author Response · Authors · 2022-11-12
> **Response to Reviewer s1nT**
>
> We thank you for the thoughtful feedback and comments. We conducted additional experiments to address the concerns raised on the amount of training needed, and we clarify specific points below. We hope that these additions address all of your concerns, though we would appreciate any additional comments or feedback that you might have.
>
> Q: Regarding our code
> We are working on open sourcing the code implementing our method. Due to the proprietary nature of the code, we will be unable to complete the process by the end of the rebuttal. Therefore, in the meantime, we have updated the appendix with pseudocode. We believe that the pseudocode is quite comprehensive. We would be happy to address any further questions about our code.
>
> **1. “how much training is required for CSIvA to obtain the results it does.”**
>
> All of our models are trained for 500k iterations. All models (whether trained on smaller or larger graphs) have converged by 350k.
>
> **2.”How much synthetic data must be generated”.**
>
> We thank you for the thoughtful question. Currently, all models are trained with 20k training samples (datasets) regardless of size of the graphs.
> However, we conducted additional experiments to better understand how much synthetic data is needed for different graph sizes. We trained our model on Dirichlet data. The number of training samples varied from 5k to 20k (5k, 10k, 15k and 20k) for different graph sizes (N=10, N=25 and N=50). We report test accuracy in SHD in the table below.
>
> | Datasets      | I=5000 | I=10000| I=15000| I=20000|
> | ----------- | ----------- |----------- |----------- |----------- |
> | N=10, ER=1 | 1.5 | 1.1  | 1.0 | 0.8  |
> | N=10, ER=2  | 3.2 | 2.2  | 1.7  | 1.4   |
> | N=25, ER=1  | 6.9 | 2.8 | 1.7| 1.4 |
> | N=25, ER=2  | 13.5 | 7.2 | 5.8 | 5.2  |
> | N=50, ER=1  | 32.0 | 12.3  | 11.3 | 8.9  |
> | N=50, ER=2  | 42.0 | 30.2 | 23.3 | 17.3 |
>
> For smaller graphs (N<=25), there is a small improvement for using more than 10k training cases. For larger graphs (N > 25) the performance consistently improves as the model uses more training data.
>
> **3. How long does the computation take and how does it compare to baseline models.**
>
> All models are trained for 500k iterations and all models have converged by then (in fact they have all converged by 400k). Once the model is trained, evaluation/inference is fast.  All models are evaluated on 128 test datasets (graphs).
>
> As we emphasize in our general remarks, the focus of causal discovery is on high-stakes test cases where accuracy matters more than compute. All models (baselines and our model) are trained on GPU. For baseline methods DCDI, each test case can take up to 48 hours to compute; baseline model ENCO takes up to 30 minutes to compute. In contrast, our method can take up to 8 hours to train for large graphs, and then inference takes minutes. (More details are provided below.) The training time of our model can be amortized if the model can be used on multiple test cases; in contrast, training time for ENCO and DCDI is proportional to the number of test cases.
>
> For our model CSIvA, training on smaller graphs (N<=20) converges within 50k, which takes up to 45 minutes. Medium sized graphs (20 < N <= 50) converge within 250k and take 4 hours. Large graphs (50<N<80) converge within 400k and take 8 hours. Note that once the model is trained, it is very fast to evaluate on different test cases. Our model is evaluated on 128 test cases.
>
> For the baseline model DCDI takes up to 4 hours on a single graph of size up to 20 (N < 20). The model takes up to 12 hours on a medium size graph (N < 50) and 48 hours for larger graphs ( 50<N<80). For the baseline model ENCO. Smaller graphs (N < 20) take 20 minutes per graph and larger graphs (N<80) take 30 minutes per graph. To compute 128 test cases, ENCO takes 2.5 days of computation time.
>
> **4. The method does not enforce acyclicity of the resulting graph.**
>
> We have incorporated your feedback and implemented the post-processing method in ENCO [1] that enforces acyclicity. We found that this post-processing did not improve the performance of our model, as our model outputs acyclic graphs.
>
> **5. Figure 4 does not seem well-presented.**
>
> We thank the reviewer for the detailed feedback. We have updated Figure 4 in our paper to group ER-1 graphs and ER-2 graphs separately.
>
> **6. Table 1 seems limited.**
>
> We thank you for pointing this out. As we are rather limited on space, the table includes methods that are the closest to our, namely supervised methods. We discussed unsupervised methods in related works. We will add a similar table including unsupervised methods we compared to in the Appendix.
>
> **7. BnLearn" should be "bnlearn", inconsistent spelling and inconsistent capitalization.**
>
> We thank the review for pointing this out. We have  edited our paper to reflect this change.
>
> 1. Efficient Neural Causal Discovery without Acyclicity Constraints. ICLR 2021. Lippe, P., Cohen, T., & Gavves, E.

---

> > ### Comment · Reviewer_s1nT · 2022-11-17
> > **Timing information added to main paper?**
> >
> > Thank you for your detailed responses, and apologies for not noticing the table in A3.  I am satisfied with what you have said, and will raise my score to 8 if you can put a summary of the information under 2 and 3 into the main paper (or at the very least into the supplement, with a clear pointer to the information from the paper).
> >
> > I also can't see any reference to the number of iterations used other than in the appendix, so again putting this in the main body would be helpful.
> >
> > ===
> >
> > One very minor other point, why not write $10^{-4}$ rather than 1e$-$4 for the learning rate?

---

> > > ### Author Response · Authors · 2022-11-17
> > > **Updated paper and thank you**
> > >
> > > Thank you for reading our rebuttal so promptly, and we are grateful that you are satisfied with our responses. We very much appreciate your time and thoughtful feedback.
> > >
> > > We have updated the main paper (and the Appendix) to reflect points 2 and 3 in our earlier response. Specifically, we included the number of iterations at the end of the first paragraph in Section 6 on page 6 and in Section 6.4 (under “Computation time”). We also included a summary of “Amount of training cases” and “computation time” in Section 6.4 of the main paper. We added the detailed discussions for the “number of training cases” in Appendix A.7.6 and “computation time we” in Appendix A.7.9. Many thanks again.

---

> > > > ### Comment · Reviewer_s1nT · 2022-11-18
> > > > **Thanks**
> > > >
> > > > Great, I'm updating my review now.

---

> ### Author Response · Authors · 2022-11-15
> **Pseudocode uploaded**
>
> We thank the reviewer again for their valuable feedback. We have uploaded the latest version of our paper.
>
> Detailed pseudocode of our model is included in Section A.8 (P31-33). We hope this has addressed the reviewer's concerns. We would appreciate any additional comments or feedbacks the reviewer might have.

---

### Official Review · Reviewer_iFSS · 2022-10-23

**Confidence:** 3
**Correctness:** 2
**Technical Novelty And Significance:** 2
**Empirical Novelty And Significance:** 3
**Recommendation:** 6

**Clarity, Quality, Novelty And Reproducibility:**

The paper writing is really good, and I only have a few questions.



- The only major concern of mine is regarding the transferability across different causal discovery tasks: they may have different graph types, different causal mechanisms, different data types, etc. This is unlike NLP or CV big models. In fact, authors of [1] also use pretraining transformer based models but the output of pretrained models is only used a good initialization.

- In the experiment, the training dataset consists of ER graphs. For out-of-distribution test, there are no results on scale-free graph (as far as I checked). Can you add experiments on this part?

- Suggestion: Transformer based NN model for causal discovery has been used in existing works in [1,2]. While presenting the used architecture in this paper, I think it beneficial to clearly state the novel part compared with [1,2].



[1] Ordering-based causal discovery with reinforcement learning.

[2] Causal discovery with reinforcement learning

**Strength And Weaknesses:**

Pros:

- a new design of transformer model to enable supervised learning for structure learning
- extensive empirical results
- writing is good and clear.

Cons:

- It is not clear how one can trust the transferability result. That is, if we can give a new setting, how accurate can we say about the output? Unlike NLP or CV tasks, I don't see what could be transferred across causal discovery tasks.



**Summary Of The Paper:**

This paper proposes a transformer based neural network architecture to learn the mapping from both observational and interventional data to graph structures in a supervised manner. The learned model is shown to be able to generalize to new synthetic graphs and is robust to some train-test distribution shifts.



**Summary Of The Review:**

I am not convinced that there is a transferrable quantity across various causal discovery tasks. I look forward to author response on this point and would like to change score if this concern is addressed.

==== after reading author response====

I am accepting the transferable part. Based on my understanding of the transferability and the listed examples, I feel that the application scope is weakened. For instance, what if we are given a totally new dataset that we do not have such prior knowledge, or that the associated prior knowledge is far different from the training datasets? I suggest authors to add more examples where the proposed method could be applied.

Due to the above point, I decide to increase my score to WA, but I also feel OK if the paper is not accepted.

---

> ### Author Response · Authors · 2022-11-12
> **Response to reviewer iFSS**
>
> We thank the reviewer for their feedback and comments. We conducted additional experiments to address reviewer's concerns about transferability, and we also clarified specific points below.
>
> **1. Regarding “Results on Scale-Free graphs (In-distribution experiments)”.**
>
> We have run additional experiments for our model on scale-free graphs, using a setting similar to [1]. We  considered scale-free graphs of degrees 4 and 6 and of size up to 50 (N <=50 ). We ran our model both on continuous and discrete valued data. For continuous value data, we ran our model on non-linear non-additive noise (NN) data. For discrete data, we ran our model on Dirichlet data with alpha = 0.1. We compared our model to the strongest baselines: DCDI and ENCO.
>
> The results on NN data are:
>
> | Dataset      | All absent | DCDI | ENCO| CSIvA|
> | ----------- | ----------- |----------- |----------- |----------- |
> | N=10, SF=4  | 20  | 4.7  | 2.7 |0.9  |
> | N=10, SF=6  | 30  | 6.8  | 4.4 |1.0  |
> | N=20, SF=4  | 40  | 9.2  | 6.5 |1.5  |
> | N=20, SF=6  | 60  | 13.1  | 8.7 | 3.4  |
> | N=30, SF=4  | 80  | 13.7  | 9.1 | 1.8  |
> | N=30, SF=6  | 120  | 19.4  | 14.7 | 6.4 |
> | N=50, SF=4  | 100  | 25.5  | 19.1 | 6.9  |
> | N=50, SF=6  | 150  | 40.1  | 29.3 | 19.2  |
>
> The results for Dirichlet data are given below
>
> | Dataset      | All absent | DCDI | ENCO| CSIvA|
> | ----------- | ----------- |----------- |----------- |----------- |
> | N=10, SF=4  | 20  | 5.1  | 3.1 |0.7  |
> | N=10, SF=6  | 30  | 7.6 | 5.2 | 1.7  |
> | N=20, SF=4  | 40  | 11.7  | 8.1 |3.1  |
> | N=20, SF=6  | 60  | 13.1  | 9.3 | 3.2  |
> | N=30, SF=4  | 80  | 17.4  | 12.7 | 4.9  |
> | N=30, SF=6  | 120  | 21.2 | 17.7 | 9.1 |
> | N=50, SF=4  | 100  | 68.1 | 43.7 | 25.8  |
> | N=50, SF=6  | 150  | 70.6  | 47.1 | 30.3  |
>
> **2. Regarding “Transferability Results (OOD on Scale-Free (SF) graphs)”.**
>
> We ran additional experiments to address reviewer’s concern regarding transferability. We trained our model on ER graphs and evaluated our model on SF graphs. Our experiments have not completed running, we will update the results within the next 24 hours.
>
> **3. Regarding comparisons to pretrained NLP and CV models.**
>
> We appreciate your concern. We would like to point out that our scenario is different from the one commonly considered in NLP and CV. In the common NLP and CV scenario, one trains a single model that needs to handle a wide variety of test data, because training the model is much more expensive and the individual test examples are lower stakes. In contrast, in our scenario training is customized to what is known about an individual high-stakes test example (for example, the understanding of gene regulation in genomics can be of much higher stake than recognizing images in Imagenet). And the task is to try to match the critical properties of the (synthetic) training data to the presumed properties of the test data (for example, in genomics, we often have expert knowledge on the type of gene regulation network graphs that the model will likely to have, we often also have an understandings of the roles of particular genes). We show that we obtain strong results whether there is a close match or less-close match (see Tables 19, 20, 21 in the paper and also the results in the 2 tables above). In contrast to unsupervised causal discovery methods (e.g., DCDI, ENCO, non-linear ICP, DAG-GNN), our trained model can be applied to multiple test cases that have similar characteristics, so it is an intermediate between the traditional causal discovery "one model per test case" and the traditional DL "one model for all test cases".
>
> **4. Regarding related papers.**
>
> We thank the reviewer for pointing us to the papers. We have added discussions of these papers in the related works section in our paper. We will upload the updated version of our paper to Openreview shortly.

---

> > ### Author Response · Authors · 2022-11-14
> > **Transferrability results**
> >
> > **Regarding “Transferability Results (OOD on Scale-Free (SF) graphs)”**
> >
> > We ran additional experiments to address the reviewer's concern regarding transferability. We trained our model on ER graphs and evaluated our model on SF graphs. We train models on ER-2, ER-3 and ER-4 graphs. For each such model, we test them on SF-2, SF-3 and SF-4 graphs.  We run our model on Dirichlet data with alpha = 0.1 and N = 10 and 20. We also report the performance of the strongest baselines DCDI and ENCO in the rightmost columns. The results are reported below.
> >
> > We find that the results from our model are relatively insensitive to the sparsity of the training graphs. And critically, we find that even training and test distributions do not match, our model still outperforms ENCO and DCDI in all training and test pairs. The benefit of our model grows as the graphs become more challenging  larger and denser). Did we suitably respond to your question?
> >
> > | Train     	| N=10	|Our model ER-2 training	|Our model ER-3 training	| Our model ER-4 training	| DCDI | ENCO
> > | ----------- | ----------- |----------- |----------- |----------- |----------- |----------- |
> > |  		| SF-2| 1.0	| 1.1 | 1.5 | 1.9 | 1.6 |
> > |  	Test	| SF-3| 1.5	| 1.7 |  1.7 | 2.8 | 2.1 |
> > |  		| SF-4| 1.4	| 1.5 | 1.5  |5.3 | 3.4 |
> >
> >
> > | Train     	| N=20	|Our model ER-2 training	|Our model ER-3 training	| Our model ER-4 training	| DCDI | ENCO
> > | ----------- | ----------- |----------- |----------- |----------- |----------- |----------- |
> > |  		| SF-2| 3.7	| 3.8 | 5.2 | 7.9 | 5.2 |
> > |  	Test	| SF-3| 7.6	| 6.3 | 7.0 | 10.6 | 8.1 |
> > |  		| SF-4| 8.5	| 7.1 | 7.4  | 11.4 | 8.9 |

---

> ### Author Response · Authors · 2022-11-18
> **Updated paper and further feedback**
>
> We thank you for your valuable feedback and suggestions. The experiments (on scale-free graphs and additional OOD experiments) you had suggested helped to strengthen our paper. We have updated our paper with these results. We appreciate your feedback. As the rebuttal time is coming to an end, we would like to ask if our reply has addressed your concern ? We would be happy to clarify any further questions.

---

> > ### Comment · Reviewer_iFSS · 2022-11-20
> > **Thanks for response and clarifications**
> >
> > Thanks very much for the response and clarifications.
> >
> > I am accepting the transferable part. Regarding "*...we often have expert knowledge on the type of gene regulation network graphs that the model will likely to have, we often also have an understandings of the roles of particular genes...*" and my understanding of the transferability, I feel that the application scope is weakened. For instance, what if we are given a totally new dataset that we do not have such prior knowledge, or that the associated prior knowledge is far different from the training datasets? I suggest authors to add more examples where the proposed method could be applied.
> >
> > Due to the above point, I decide to increase my score to WA, but I also feel OK if the paper is not accepted.

---

> > > ### Author Response · Authors · 2022-12-06
> > > **Thank you and transfer learning**
> > >
> > > We thank the reviewer for reading our rebuttal, and we very much appreciate the reviewer's time and thoughtful feedback. We are grateful that the reviewer has revised their score.
> > >
> > > Regarding "For instance, what if we are given a totally new dataset that we do not have such prior knowledge, or that the associated prior knowledge is far different from the training datasets?"
> > >
> > > We understand the reviewer's concern. This is a very good question. Our experimental results suggest that the proposed method is robust to the training data (section 6.2 of the paper). We have trained and evaluated our model on different graph sparsity and different conditional probability distributions. The model performed well despite the mismatch between training and test datasets.
> > >
> > > When there is no prior knowledge about the graph, we can train our model on synthetic data that generalized well OOD. This was the case for the experiments in section 6.3, where the model is trained on synthetic data with no prior knowledge about the real world datasets.  The hyperparameters for training data were chosen aprior. Models were trained on synthetic graphs with sparsity of ER-2 and α = 0.25, as we found in section 6.2 that these hyerparameters worked well overall and hence were chosen. Our model significantly outperformed all other baselines while having no prior knowledge about the test datasets.
> > >
> > > Then we also conducted additional experiments as the reviewer had suggested. We trained and tested our model on different graph models and  different graph sparsities simultaneously. we trained our model on ER graphs and evaluated our model on SF graphs with average edge degree of 2, 3 and 4.  To be specific, we train models on ER-2, ER-3 and ER-4 graphs. For each such model, we test them on SF-2, SF-3 and SF-4 graphs.  We run our model on Dirichlet data with alpha = 0.1 and N = 10 and 20. We also report the performance of the strongest baselines DCDI and ENCO in the rightmost columns. The results are reported below. Our model outperformed DCDI and ENCO despite having mismatches between training and test datasets.
> > >
> > > Overall, we find that the performance of our model are relatively insensitive to various hyperparameters of the training data, including graph sparisty, type of graph model being used, as well as conditional probability distributions. And critically, we find that even training and test distributions do not match, our model still outperforms ENCO and DCDI in all training and test dataset pairs.
> > >
> > > | Train     | N=10 |Our model ER-2 training |Our model ER-3 training | Our model ER-4 training | DCDI | ENCO
> > > | ----------- | ----------- |----------- |----------- |----------- |----------- |----------- |
> > > |   | SF-2| 1.0 | 1.1 | 1.5 | 1.9 | 1.6 |
> > > |   Test | SF-3| 1.5 | 1.7 |  1.7 | 2.8 | 2.1 |
> > > |   | SF-4| 1.4 | 1.5 | 1.5  |5.3 | 3.4 |
> > >
> > >
> > > | Train     | N=20 |Our model ER-2 training |Our model ER-3 training | Our model ER-4 training | DCDI | ENCO
> > > | ----------- | ----------- |----------- |----------- |----------- |----------- |----------- |
> > > |   | SF-2| 3.7 | 3.8 | 5.2 | 7.9 | 5.2 |
> > > |   Test | SF-3| 7.6 | 6.3 | 7.0 | 10.6 | 8.1 |
> > > |   | SF-4| 8.5 | 7.1 | 7.4  | 11.4 | 8.9 |
> > >
> > > We hope this additional information clarifies our findings and their significance. Thank you again for your valuable feedback.

---

### Official Review · Reviewer_uyeG · 2022-10-24

**Confidence:** 4
**Correctness:** 3
**Technical Novelty And Significance:** 2
**Empirical Novelty And Significance:** 2
**Recommendation:** 5

**Clarity, Quality, Novelty And Reproducibility:**

# Clarity

The paper is well-written and easy to follow.

# Quality

The figures could be far better; they are hard to read (even with a very large screen), are clearly not vector based (have a look at IPEDraw) and are in general of sub-par quality compared to the rest of the graphs.

# Novelty

The novelty is low. The authors have combined known parts into a new network which has some interesting properties but no theoretical results which show that this is a general method which would work in all settings. Now, this is not to absolve their competitor methods of this same analysis, they are under the same critique, but this paper is the one under scrutiny at the moment.

# Reproducibility

I imagine the authors will release their model in due course hence there are not concerns in this area.

**Strength And Weaknesses:**

This is the main review. It is organised according to the headings of your paper.

# Abstract

- First sentence, perhaps we can be a bit less specific and say that graph learning is _one_ fundamental problem in causal inference (there are many).
- First question; if yours is a supervised approach, what happens when you have a setting for which you have no data? Learning directed acyclic graphs (DAGs) from data is an NP-hard problem after all.
- It would help if you in the appendix stated if your method assumed causal sufficiency or not.

# Introduction

- Excellent first paragraph.
- You say: "Our approach can be viewed as a form of meta-learning, as the model learns about the relationship between datasets and structures underlying them" - what precisely is "meta" about this?
- What is a "naturalistic CBNs"? A commonly occurring CBN, something we would find in nature?
- Please make the subfigures larger in figure 1; even with an exceptionally large screen, it is difficult to read the text in your images.
- I am curious: what do you do about data scarcity and imbalance (the question relates even more to the latter)? There is very little interventional data knocking about, and when there is, there is usually not very much of it compared to the amount of observational data in the same setting which would suggest that your approach has to deal with the data imbalance problem too. How do you do that?
- Again... what is a "naturalistic" CBN?

# BACKGROUND

- If you are tight on space, you can reduce both of these sections and just link to the relevant literature. You've given the main sources for CBNs and a few sentences are enough IF you are short on space. If not, leave as is, it is good.

# CAUSAL STRUCTURE INDUCTION VIA ATTENTION (CSIVA)

- The last paragraph on page three could do with a few equation blocks; it is currently very dense and hard to follow.
- I cannot help but think that section 3.2 could have been considerably shorter by simply employing a depiction (image) of the encoder and decoder and simple explanations to go alongside it. This long and verbose text to explain a model seems far too verbose when there is no novelty in the individual parts, simply just how they are put together - which could be more easily shown with an image.

# SYNTHETIC DATA

- Careful not to confuse your reader when you discuss identifiability, as it has two meanings within the field of causal inference (and I have not seen it used the way you are using it). See Pearl, 2009 for a formal take - essentially we say that an intervention is 'identifiable' if it can be composed of observational distributions i.e. $p(y| do(x)) = p(x|y)p(x)$. The way Eberhardt et al. use the word is not common and not standard and clashes with a much more central domain within causal inference as noted.

# EXPERIMENTS

- Again, I think you are being far too verbose in your writing. Figure 3 clearly tells us what we need to know regarding the in-distribution experiments, you do not need separate paragraphs for each type of data.
- It would be helpful if you had included some graph topologies which CSiVA actually learned compared to the other methods.
- When did you stop learning? Table 3 only tells us the hyperparameters but not when you consider a graph 'learned' from CSiVA's perspective.

# DISCUSSION

- It would have been more useful if you had shown that this method is useful in an RL setting in which an RL agent interacts with the environment via observing low-level pixel data.



**Summary Of The Paper:**

This paper presents a novel deep-learning model capable of inferring the underlying causal structure. The method does this by treating the process as a black-box function which is estimated through a network that maps observational and interventional data to a causal diagram (the output). They demonstrate their approach in a number of experimental settings.

**Summary Of The Review:**

The authors have contributed a novel deep-learning model which has some interesting properties when it comes to causal discovery. They demonstrate impressive empirical results on a range of datasets but ultimately contribute very little methodological or theoretical novelty. Their contribution is empirical which is examined in detail in the review.

---

> ### Author Response · Authors · 2022-11-11
> **Response to Reviewer uyeG**
>
> We are grateful for the reviewer’s detailed feedback and comments. We have updated our paper to reflect the changes the reviewer suggested and we will upload the updated version on Openreview soon. We ran additional experiments to address reviewer’s questions about data imbalance and scarcity, and we also clarified specific points about novelty below.
>
> 1. **Regarding “naturalistic CBNs”**
>
> By “naturalistic”, we are referring to the non-synthetic nature of the graphs taken from the bnlearn repository, in the sense that these graphs correspond to causal relationships that exist in nature. For example, the Sachs graph represents the causal pathways linking several proteins in human immune system cells as agreed upon by experts. We will add this clarification to the introduction.
>
> 2. **Regarding “meta-learning”**
>
> We have clarified this sentence in the introduction.
>
> 3. **Regarding “data scarcity and imbalance ”**
>
> As the reviewer notes, interventional data is often scarce (though not in all domains). To optimize our model’s performance in an actual use case, we would of course match the proportion of interventional data in our synthetic training set to the proportion available for the test case. However, to address the reviewer’s implicit concern that a large amount of interventional data is necessary for the model’s success, we evaluated our model on a variable proportion of interventions in section A.7.3 and figure 7. We varied the amount of interventions in the data from 0% (only observational data) to 100% (only interventional data). We can see that the model’s performance improves continuously as it observes more interventional data.
> Another possible concern regarding interventions is that only certain nodes can be intervened on, as is common in, say, genomics. As a result, some nodes are never intervened on in the training set, which makes inference more challenging. To address the different sort of  ‘data imbalance’ that this restriction imposes, we ran experiments on non-linear non-additive noise (NN) data graphs of size 10 with sparsity ER-1 and ER-2, varying the percentage of nodes that can be intervened on from 10% (on average only 1 node is intervened on in an entire dataset) to 100% (on average, all 10 nodes are intervened on in an entire dataset). The results are:
>
> | Number of variables      | Sparsity of graph | 10% | 30% | 50% | 70% | 90% | 100% |
> | ----------- | ----------- | ----------- | ----------- | ----------- | ----------- | ----------- | ----------- |
> | N=10   |  ER-1      | 6.4  | 5.0  | 3.6   |  2.2   | 0.9   | 0.4   |
> | N=10   |  ER-2      |11.9  | 9.5  | 7.6  | 4.8   | 3.0   | 1.7  |
>
> We observe that the performance of the model improves as more nodes can be intervened on. Note that these graphs are random graphs and the nodes to be intervened on are picked randomly.
>
> We hope this has addressed the reviewer’s question. We would be happy to discuss/ conduct any additional experiments if the reviewer has additional experiments/ concerns in mind, or if we have misinterpreted the comment.
>
> 4. **Regarding the writing**
>
> We very much appreciate the reviewer’s detailed feedback and suggestions on improvements for the writing, and we are grateful for the reviewer’s time. We have updated our paper with the reviewer’s suggestions. We will upload the updated version on Openreview soon.
>
> 5. **Regarding “when do we stop learning?”**
>
> All of our models are trained for 500k iterations. We did not need to do early-stopping, as we did not see problems of overfitting. We will update our paper with a plot of the evaluation performance over time (training iterations) for different sized graphs.
>
> 6. **Regarding “Novelty”**
>
> - (1) regarding no theoretical results: we have discussed identifiability of the data and consistency of our method in the paper. Like a lot of work in DL, the results are empirical and the promising empirical results will hopefully spur work in theoretical analysis, as it has in DL more broadly, but that is beyond the scope of our work, which aims at showing strong empirical results.
>
> - (2) while it is true that the transformer architecture is ubiquitous, the approach we have taken is quite unlike traditional uses of the transformer and quite unlike traditional approaches to causal discovery. Our approach is a meta-inference approach in which the training 'examples' are in fact synthetic data sets. Several of the authors on this paper were skeptical a priori that such an approach could work, especially with synthetic data, so it is a surprising and unexpected finding, for which we've obtained strong empirical support in order to justify publication. Another respect in which our work is distinguished from nearly all work in DL is its focus on training a model for specific test cases (see general comment above).

---

> ### Author Response · Authors · 2022-11-22
> **Updated paper and further feedback**
>
> We thank you for your detailed feedback and suggestions. Your suggestions on the writing and experiments helped us to improve our paper: we have added the results on imbalance data and also updated Figure 1. We would like to ask if our reply and updates have addressed your concern? We would be happy to clarify any further questions.

---

> > ### Author Response · Authors · 2022-12-04
> > **Feedback**
> >
> > Dear Reviewer,
> >
> > Thank you again for your thoughtful review. Does our response address your questions? We would appreciate the opportunity to engage further if needed.
> >
> > Thanks for your time.
> >
> > Many thanks,

---

> ### Author Response · Authors · 2022-12-12
> **Feedback: End of Discussion Period**
>
> Dear Reviewer,
>
> Thank you again for your thoughtful review. Does our response address your questions? We would appreciate the opportunity to engage further if needed.
>
> Thanks for your time.
>
> Many thanks,

---

### Official Review · Reviewer_REpm · 2022-10-28

**Confidence:** 4
**Correctness:** 2
**Technical Novelty And Significance:** 3
**Empirical Novelty And Significance:** 2
**Recommendation:** 5

**Clarity, Quality, Novelty And Reproducibility:**

## The proof of consistency and identifiability for the proposed method

Sec. 3 mentions consistency, but please elaborate on it.
"The data-sampling distribution t(G, D) and the MLE objective uniquely determine the target distribution learned by the model."

Sec. 4 mentions the identifiability, please elaborate on the proposed method.
"As discussed in Eberhardt et al. (2006), in the limit of an infinite amount of such single-node interventional data samples, Gi is identifiable."

Please elaborate and prove
1. what is the target distribution?
2. why the target distribution can be uniquely determined?
3. the consistency of the results.
4. the identifiability of the proposed method



## About In and out of distribution
I have a question about the experiment setting. As for causal discovery, there is only one graph per application for testing. So what does it mean in or out of distribution? Is it all about creating training data sets such that ( for example ) the in-distribution setting requires that the training data cover the parameters for generating the data of the underlying causal graph in an application?

Furthermore, if the ID and OOD settings only differ according to covering the statistical parameter values for generating testing data or not (like the graph density and the parameter values of Dirichlet prior), then this can over-simplify the OOD case. Because as long as creating the training data in a proper way, even though the parameters are not exactly the same as the testing case, by interpolation or extrapolation, it should easily recover the test graph. Besides, in most cases, the mechanisms which shift away from ID and training data are unknown.


Therefore, if this is the case, the paper may over-claim its contribution regarding this point.



**Strength And Weaknesses:**

__Strength__

It deserves to encourage different thoughts and problem formulations for causal discovery. The idea of leveraging synthetic data,  supervised learning, and meta-learning for solving causal discovery problems is interesting and new. If there are related works with similar ideas that I may not realize, then the novelty should be deducted; otherwise, I would like to support such new ideas in causal discovery. However, the work still requires more rigorous considerations.


__Weakness__

The work proposes a framework for causal discovery. However, besides the accuracy, causal discovery also cares about what are the learned results, what are their properties, and what guarantee we have. For example, we know that supervised learning can suffer from
 covariate shift, poor transfer learning performance, the lack of uncertainty, overfitting, etc al. Then how would they bother the proposed method? Maybe the paper could discuss such points. It mentions OOD performance of which I will provide detailed comments in the clarification part.

Moreover, some other aspects can be interesting for causal discovery is that
1. Can you show the identifiability and consistency of the method? (I will further explain this point in the clarification part.)
2. Are the results in Markov equivalent class of the underlying causal graph?
3. How are the results related to the existing results about causal discovery, e.g., the identifiability of different functional causal models (ANMs, PNLs, etc al.)?
4. Try to explain how the discrepancy between synthetic/simulation and real-world data is or is not a problem for the method. And what can be the problem? Because it is well-known that the evaluation of causal discovery on synthetic data is too simple compared with real-world data. Then why such a gap between synthetic data and real-world data is (or not) a problem for the proposed method?

__Some minor concerns:__

In Sec, 1, it mentions that the work is related to meta-learning. However, the method doesn't seem to have any meta-learning module to deal with the distribution shift or OOD problem.
In Sec. 3.1
1. Why distributions are denoted by t, not the convention p?
2. Why there are $N^2$ numbers of $A_l$, for DAGs should it be less ?


**Summary Of The Paper:**

The paper proposes an interesting supervised learning setting for causal discovery. Firstly, it creates synthetic training data ( observational and international data ) with various simulators and settings. And then it designs a neural network that outputs the causal graphs. Based on such a supervised learning manner, it claims that the method can recover the ground-truth graph (which lacks analysis and proofs, see details in the following comments) and can handle the distribution-shift / out-of-distribution problem (which can be over-claiming the contribution, see details in the following comments).

**Summary Of The Review:**

The new and interesting idea of causal discovery deserves support and encouragement, but the paper requires more consideration towards causal discovery.

---

> ### Author Response · Authors · 2022-11-16
> **Response to reviewer REpm (1/2)**
>
> We thank the reviewer for the valuable feedback, and for the encouragement of our novel approach to think about causal discovery. We hope to address the concerns below.
>
> **1. Issues with supervised learning (covariate shift, poor transfer learning performance,  lack of uncertainty)**
>
> While supervised learning suffers indeed from these issues, we believe and prove empirically that those issues are less severe in our method because of the nature of our approach. More precisely, this  is because we do not build a single model for all test cases, but rather separate models, with each model customized to a high-valued test scenario. Of course, we are not guaranteed that the assumptions we make about the test scenario are captured by the synthetic training dataset, but we have a greater opportunity to match than is the case of ordinary supervised learning tasks (e.g., imagenet). Nonetheless, we do extensive empirical investigation of mismatch between training and testing scenarios (covariate shift), as we report in Tables 17, 18, 19, 20 and 21 and in the discussion in Supplementary A.6.
>
> **2. Are the results in Markov equivalent class of the underlying causal graph?**
>
> We randomly chose 24 test samples that our model has generated (with different graph sparsity).  The generated graphs are shown in Figures 7 and 8, the graphs that differ from the ground truth graphs (4 out of 24) are not always in the same Markov Equivalence class of the ground truth graphs. Specifically 3 out of 24 graphs that were generated are not in the same Markov equivalence class. However, the generated graphs on average only get 1 or 2 edges incorrectly.
>
> **3. Try to explain how the discrepancy between synthetic/simulation and real-world data is or is not a problem for the method. And what can be the problem? Because it is well-known that the evaluation of causal discovery on synthetic data is too simple compared with real-world data. Then why such a gap between synthetic data and real-world data is (or not) a problem for the proposed method?**
>
> The key hypothesis that we investigated was whether synthetic data could be used for training, and we obtained strong positive results.  One reason why synthetic data might be appropriate here but not in, say, image tasks, is because only certain high-level properties of the inferred graphs are relevant to causal discovery; these properties include graph sparsity, conditional probability distributions, etc. We are able to manipulate these high-level properties in our data-generative processes. In contrast, in the case of images, there exists no simple generative model that characterizes what makes an individual pixel 'lifelike' (realistic).

---

> ### Author Response · Authors · 2022-11-16
> **Response to reviewer REpm (2/2)**
>
> **4. Regarding target distribution, consistency and identifiability of our method.**
>
> The data-generating process described at page 5 corresponds to ancestral sampling of a joint distribution over dataset ${\mathcal D}$ and graph ${\mathcal G}$. We denoted this joint distribution as $t({\mathcal G}, {\mathcal D})$.
>
> The target distribution is the posterior distribution $t({\mathcal G}|{\mathcal D})\propto p({\mathcal D}|{\mathcal G})t({\mathcal G})$.
> The likelihood $p({\mathcal D}|{\mathcal G})$ is defined as $p({\mathcal D}|{\mathcal G}):=\prod_{s=1}^S p^s(X_1=x^s_1,\ldots,X_N=x^s_N)$, where $p^s(X_1=x^s_1,\ldots,X_N=x^s_N)$ is the observational distribution over the nodes $X_1,\ldots,X_N$ if $x^s_1,\ldots,x^s_N$ is an observational data sample, or the interventional distribution if $x^s_1,\ldots,x^s_N$ is an interventional data sample (see the third line at beginning of page 3 for a description of the interventional distribution).
>
> The target distribution $t({\mathcal G}|{\mathcal D})$ is uniquely and well defined by the likelihood and the prior $t({\mathcal G})$.
>
> Unlike unsupervised methods (score-based or constraint-based), we directly train our model to predict the graph. Hence, the MLE objective used to train our model, $L = \mathbb{E}_{t({\mathcal G}, {\mathcal D})}[\ln \hat{t}({\mathcal G}|{\mathcal D})]$ is equivalent to minimizing the KL-divergence $KL(t({\mathcal G}|{\mathcal D}); \hat{t}({\mathcal G}|{\mathcal D}))$ between the target posterior distribution and the model's distribution $\hat{t}({\mathcal G}|{\mathcal D})$.
>
> The KL-divergence has an unique global minimum, which is when the two distributions are equal almost everywhere. In the infinite model capacity limit, where the target distribution is contained in the set of distributions that can be realized by the model, we should expect the model to converge to the ground-truth posterior distribution $t({\mathcal G}|{\mathcal D})$.
>
> Identifiability in our case translates into the possibility for the posterior to become concentrated around the true graph. This is always true in our case, since our datasets contain random interventions on all nodes: the theoretical results in Eberhardt 2006 state that, with N-1 single-node hard interventions, the causal relationships between N nodes can be determined. This is true with perfect data, which translates into a sufficient amount of observation for each intervention that represents the underlying distribution.
>
> The reason why we used t is to distinguish the notations for the distribution of the nodes and the distribution of the graph, but we realized that this might be confusing, therefore we will change $t$ back into $p$ in the next version.

---

> ### Author Response · Authors · 2022-11-29
> **Further feedback**
>
> We thank you for your detailed feedback and suggestions. Your suggestions helped us to improve our paper: we have clarified Issues with supervised learning, markov equivalence class and target distribution, consistency and identifiability. We would like to ask if our reply and updates have addressed your concern? We would be happy to clarify any further questions.
>
> We also note that we ran more experiments clarifying concerns raised by other reviewers, and summary of the experimental results can be found under the heading "Summary of updates". Thank you for your time and feedback. We appreciate it.

---

> ### Author Response · Authors · 2022-12-06
> **Futher feedback, end of discussion period.**
>
> Dear Reviewer,
>
> Thank you again for your thoughtful feedback. Does our response address your concerns? As the discussion period is ending soon, we'd appreciate any additional thoughts or concerns about our work.
>
> Many thanks

---

### Official Review · Reviewer_UgAy · 2022-11-03

**Confidence:** 4
**Clarity, Quality, Novelty And Reproducibility:** The clarity, quality, novelty and rep…
**Correctness:** 3
**Technical Novelty And Significance:** 4
**Empirical Novelty And Significance:** 4
**Recommendation:** 8

**Strength And Weaknesses:**

– Of the four contributions listed at the end of Section 1, only the last two are contributions. Introducing approaches and architectures is easy, but showing whether, how, and why they work is difficult.

– The authors use the term "causal Bayesian network". While the terminology of causal graphical models is hardly unified, a more widely used and accurate term would be either "causal graphical model," "structural causal model," or "directed graphical model." The models targeted in this paper are not necessarily Bayesian, so "causal Bayesian network" isn't really accurate.

– The current version of the experiments uses structural Hamming distance as a measure of accuracy. The authors should strongly consider using more useful measures of structural accuracy.  Structural Hamming distance has been shown to be a poor measure of model quality. Instead, consider SID (Peters & Bühlmann 2015) or BSF (Constantinou et al. 2021).

+ Even given the use of SHD, the experimental methodology and results are impressive. The authors perform experiments under a wide range of circumstances, far wider than the typical paper. The use of a variety of methods (ablation studies, visualization, effect of interventional data, etc.) is particularly impressive. In all cases, the proposed method performs much better than existing baselines.

– Even given the impressive results, readers would benefit from a more through analysis showing what sorts of errors were corrected in CSIvA’s results and explaining why this happened. Were the edges (that were correctly added or omitted) weak or strong? Was it primarily about edge orientation or edge existence? Was it primarily about adding missing edges or omitting incorrect edges?

References

Peters, J., & Bühlmann, P. (2015). Structural intervention distance for evaluating causal graphs. Neural computation, 27(3), 771-799.

Constantinou, A. C., Liu, Y., Chobtham, K., Guo, Z., & Kitson, N. K. (2021). Large-scale empirical validation of Bayesian Network structure learning algorithms with noisy data. International Journal of Approximate Reasoning, 131, 151-188.

**Summary Of The Paper:**

The authors propose a method for learning the structure of a directed graphical model using supervised learning from synthetically generated data. They perform extensive evaluation of the proposed method, with impressive results.

**Summary Of The Review:**

The empirical results from applying the proposed method are very impressive and should not be ignored. Clearly, there is more work to do, but the existing work presented here should be published so that others can build on these results.

---

> ### Author Response · Authors · 2022-11-14
> **Response to reviewer UgAy**
>
> We appreciate the reviewer for their positive and detailed feedback. We attempt to address the questions raised by the reviewer and would be more than happy to address any additional questions that the reviewer may have.
>
> **Regarding evaluation criteria.**
>
> We thank the reviewer for pointing this out. We evaluated our model against additional evaluation metrics that the reviewer had suggested. To be specific, we report results of our model on the balanced scoring function (BSF), F1 score, precision, recall and also area under ROC (AUROC).  We run our model on both discrete and continuous data. For discrete data, we run our model on Dirichlet data with alpha = 0.1, and for continuous data, we run the model on non-linear non-additive noise model (NN) data. The results are reported below.
>
> For Dirichlet data, our model achieved strong results. Again, the area under ROC (AUROC) is greater than 0.95 for all cases. The F1 score is also always above 0.90.  For continuous data, our model achieved even stronger results. Our model scored between 0.98 and 0.99 for all scores. This is an indication that the edges were almost always correct and the correct edges also have strong probability.
>
> | Dirichlet data      |  SHD | BSF | AUROC| F1| Precision | Recall
> | ----------- | ----------- |----------- |----------- |----------- |----------- |----------- |
> | N=10, ER=1  | 0.5  | 0.91  | 0.96 | 0.95  | 0.97 | 0.93  |
> | N=10, ER=2  | 1.0  | 0.93  | 0.98 | 0.95  | 0.96 | 0.93  |
> | N=20, ER=1  | 2.4  | 0.91  | 0.98 | 0.90  | 0.88 | 0.92  |
> | N=20, ER=2  | 3.3  | 0.94  | 0.99 | 0.94  | 0.91 | 0.95  |
>
>
> | Non-linear non-additive noise data      |  SHD | BSF | AUROC| F1| Precision | Recall
> | ----------- | ----------- |----------- |----------- |----------- |----------- |----------- |
> | N=10, ER=1  | 0.17  | 0.98  | 0.99 | 0.99  | 0.99 | 0.98  |
> | N=10, ER=2  | 0.30  | 0.98  | 0.99 | 0.99  | 0.99 | 0.98  |
> | N=20, ER=1  | 0.12  | 0.99  | 0.99 | 0.99  | 0.99 | 0.99  |
> | N=20, ER=2  | 0.62  | 0.99  | 0.99 | 0.99  | 0.99 | 0.98  |

---

### Author Response · Authors · 2022-11-12
**General comment to all reviewers**

We thank all the reviewers for their time and detailed feedback on our paper. Feedback by the reviewers have been very helpful. We are also glad that the reviewers found our paper to have "impressive results" (Reviewer UgAy and Reviewer S1n1) and “...the clarity, quality, novelty and reproducibility are all well above threshold” (Reviewer UgAy).

We would like to address one point of confusion in particular, which concerns the fact that our training scenario is quite different from the standard setting in machine learning.  In the standard setting, one trains a single model then applies the model to a wide range of test cases; in this setting, training the model is expensive and the individual test examples are low stakes. In contrast, in our causal-discovery setting, we are focused on domains where there might be just a single high-stakes test example (e.g., inferring the causal structure of a gene regulatory network). Because of the nature of this challenge, we can afford to train a model that is optimized to perform well for that particular test example. Thus, we incorporate whatever knowledge we have of the test example into the training set.  Our approach matches the critical properties of the (synthetic) training data to the presumed properties of the test case. We show that we obtain strong results whether there is a close match or not (Tables 15, 16, 17). In this causal-discovery setting, previous methods—both DL and non-DL—are similarly customized to individual test cases. In contrast to many of these previous methods, our trained model can be applied to multiple test cases that have similar characteristics, so it is an intermediate between the traditional causal discovery "one model per test case" and the traditional DL "one model for all test cases".

Our “one model for a few test cases” scenario might help to alleviate reviewer concerns about the transferability of the learned model across causal discovery tasks and about whether a model trained on one data distribution would work on a test case drawn from a different distribution. Because we train a model for each particular test case (or test distribution, in the case of our experiments with synthetic data), we have the opportunity to incorporate whatever domain knowledge is available, e.g., rough knowledge of edge sparsity, graph size, the number of observational and interventional samples in a data set. We show that our model is fairly robust to mismatches even at the coarse level of a different edge sparsity in training and testing (Tables 15 and 16). However, in causal discovery scenarios, the data set typically comes with some information about graph hyperparameters which is knowledge we leverage for training.

We conducted additional experiments, and edited the paper to incorporate reviewer’s feedback and to make it more concise. We will also open source the code for our method. This will take some time due to the approval process. In the meantime, we have provided pseudo code for our model in the appendix.

---

### Author Response · Authors · 2022-11-15
**Summary of updates.**

We thank all reviewers again for their time and valuable feedback on our paper. We have uploaded an updated version of our paper. The updates are listed as follows:

**Additional evaluation metrics.**
We further analyzed the performance of our model using additional evaluation metrics, such as balanced scoring function (BSF), f1 score, precision, recall and area under ROC (AUROC). Results are reported in Tables 22 and 23 in section A.7.2 in the appendix. Our model achieved strong performance on all scores.

**Imbalanced data by limited interventions.** We conducted additional experiments to understand the performance of our model when the data is not balanced. In this case, we limited the percentage of nodes that can be intervened on in a dataset. The results are reported in Table 25 and discussed in Section A.7.7 in the appendix.

**Scale-free graphs.** We further evaluated our model on scale-free graphs. The results are reported in Tables 14 and 15 and discussed in Section A.5.2. Our model significantly outperforms strong baselined DCDI and ENCO in all cases.

**Additional OOD experiments**: We also conducted additional OOD experiments, where the model is trained and tested on different graph-types (ER and scale-free). The results are reported in Tables 20 and 21 in Section A.6.3. We also compared the performance of our model evaluated on OOD test data to the performance of baseline models DCDI and ENCO. We find that the results from our model are relatively insensitive to the sparsity of the training graphs. And critically, we find that even training and test distributions do not match, our model still outperforms ENCO and DCDI in all training and test pairs. This is another indication that our model generalizes well OOD.

**Effect of dataset size**: We conducted additional experiments to understand the effect of the dataset size used for training. Results are reported in Table 25 and discussed in Section A.7.6. For smaller graphs (N<=25), there is a small improvement for using more than 10k training cases. For larger graphs (N > 25) the performance consistently improves as the model uses more training data.

**Pseudo-code**: We have attached detailed pseudocode at the end of the appendix in section A.8, and are aiming to open-source the code by release of the camera-ready version.

---

### Decision · Program_Chairs · 2023-01-20

**Decision:**

Accept: poster

**Justification For Why Not Higher Score:**

Lack of consensus, concerns regarding identifiability and novelty

**Justification For Why Not Lower Score:**

High overall score and endorsement of several well-known experts in causal discovery and structure learning

**Metareview: Summary, Strengths And Weaknesses:**

This paper presents a controversial approach to structure learning by treating the problem as a black-box prediction task. Although some of the reviews were borderline, I concur with the enthusiasm of several expert reviewers in the field who appreciated the novelty of the overall approach. I recommend acceptance.

I hope the authors will take the many suggestions for improvement into account, and especially to make sure that fully reproducible code is made available in the camera-ready.

**Note From Pc:**

if the above contains the word "oral" or "spotlight" please see: "oral" presentation means -> notable-top-5% and "spotlight" means -> notable-top-25%. As stated in our emails, we are disassociating presentation type from AC recommendations

**Summary Of Ac-Reviewer Meeting:**

N/A